# Dynamic Online Ensembles of Basis Expansions

**Daniel Waxman**                                                       *daniel.waxman@stonybrook.edu*
*Department of Electrical and Computer Engineering*
*Stony Brook University*

**Petar M. Djurić**                                                       *petar.djuric@stonybrook.edu*
*Department of Electrical and Computer Engineering*
*Stony Brook University*

**Reviewed on OpenReview:** *https://openreview.net/forum?id=aVOzWH1Nc5*

## Abstract

Practical Bayesian learning often requires (1) online inference, (2) dynamic models, and (3) ensembling over multiple different models. Recent advances have shown how to use random feature approximations to achieve scalable, online ensembling of Gaussian processes with desirable theoretical properties and fruitful applications. One key to these methods' success is the inclusion of a random walk on the model parameters, which makes models dynamic. We show that these methods can be generalized easily to any basis expansion model and that using alternative basis expansions, such as Hilbert space Gaussian processes, often results in better performance. To simplify the process of choosing a specific basis expansion, our method's generality also allows the ensembling of several entirely different models, for example, a Gaussian process and polynomial regression. Finally, we propose a novel method to ensemble static and dynamic models together.

## 1 Introduction

Many machine learning applications require real-time, online processing of data, a setting that often necessitates significant modifications to standard methods. Online adaptations of many different methods have been derived, including kernel machines (Kivinen et al., 2004; Lu et al., 2016), (kernel) least-squares (Kay, 1993; Engel et al., 2004), and Gaussian processes (Gijsberts & Metta, 2013; Stanton et al., 2021). Online learning has also been studied extensively from the optimization perspective (Orabona, 2019; Hazan, 2022).

Online learning may be additionally complicated by the need for model selection, as it is rarely clear at the beginning of the learning process which model will perform the best. One option in this case is to train several models in parallel and *ensemble* them. In a Bayesian context, Bayesian model averaging (BMA) (Hoeting et al., 1999) has long been used to ensemble online models, and works by weighting each "expert" model by its evidence.

More recently, Lu et al. (2022) showed how to adapt BMA to online Gaussian processes (GPs) in a technique they call *incremental ensembles of GPs (IE-GPs)* (see also Lu et al. (2020)). GPs are a flexible, non-parametric tool of Bayesian machine learning that enjoys universal approximation properties and principled uncertainty estimates (Rasmussen & Williams, 2005). Through a random Fourier feature (RFF) approximation to Gaussian processes (Rahimi & Recht, 2007; Lázaro-Gredilla et al., 2010), online learning can be performed (Gijsberts & Metta, 2013), with closed-form Bayesian model averaging updates and tractable regret analysis (Lu et al., 2022).

In addition to an online ensemble of GPs, Lu et al. (2022) illustrate the benefits of allowing random walks on model parameters, which they call *dynamic IE-GPs (DIE-GPs)*. This can dramatically increase performance when the learning task changes slightly over time. For example, when tested on the SARCOS dataset

(Rasmussen & Williams, 2005), which consists of learning the inverse dynamics of a robotic arm, DIE-GPs significantly outperform IE-GPs (Lu et al., 2022).

Taking a more global view, the ensembling of random feature GPs, as introduced by IE-GPs, has proven flexible and effective; extensions to the framework include Gaussian process state-space models (Liu et al., 2022), deep Gaussian processes (Liu et al., 2023a), and graph learning (Polyzos et al., 2021). Together with its extensions, DIE-GPs have been successfully used for Bayesian optimization (Lu et al., 2023) and causal inference (Liu et al., 2023b).

However, reliance on the RFF approximation means that IE-GPs inherit the weaknesses of random feature GPs. Namely, the RFF approximation is a direct Monte Carlo approximation of the Wiener-Khinchin integral and therefore suffers harshly from the curse of dimensionality. We show in our results that on several real-world datasets, (D)IE-GPs provide similar or worse performance than simpler models, including online Bayesian linear regression and one-layer RBF networks.

In this paper, we introduce *online ensembles of basis expansions (OEBEs)*, a generalization of IE-GPs that break their reliance on RFF GPs and improve performance over several real datasets. In particular, we make the following contributions:

1. We note that the derivation of DIE-GPs does not rely at all on the RFF approximation other than it being a linear basis expansion. In fact, the same derivations and code can be reused to ensemble arbitrary Bayesian linear models with any design matrix. This allows not only the ensembling of the same type of model together, but also the ensembling of many different basis expansions (e.g. B-splines, one-layer RBF networks, and more).

2. We argue that, if GP regression is of interest, a GP with a generalized additive model (GAM) structure is often more appropriate. To this end, we use GAM Hilbert space Gaussian processes (HSGPs) (Solin & Särkkä, 2020), which can be viewed as a quadrature rule of the same integral that the RFF approximation targets by direct Monte Carlo. Besides theoretical arguments, empirical results suggest HSGPs converge to the true approximated GP more quickly (in terms of the number of basis functions) than RFF GPs (Riutort-Mayol et al., 2023). We provide a similar empirical analysis.

3. We introduce a novel method to combine static and dynamic models, enabling the use of principled posteriors of static methods when appropriate and expanding expressiveness of dynamic methods otherwise. We show that this method is necessary by providing a constructive example on real data where the naïve approach to ensembling static and dynamic methods fails.

4. We provide Jax/Objax (Bradbury et al., 2018; Objax Developers, 2020) code at `https://www.github.com/danwaxman/DynamicOnlineBasisExpansions` that only requires the user to specify the design matrix, with several choices already implemented.

The rest of this paper is structured as follows: in Section 2, we review background concepts in linear basis expansions, GP regression, spectral approximations of GPs, and BMA. These concepts are operationalized in Section 3, where we introduce the OEBEs and several extensions, including applications to non-Gaussian likelihoods, and offer some brief theoretical remarks. We provide additional practical comments regarding the construction of OEBEs, including discussion on the contents of an ensemble and how to ensemble static and dynamic models in Section 4. The proposed models are tested empirically in Section 5. Finally, we offer some concluding remarks and future directions in Section 6.

## 2 Background

In this section, we begin by reviewing Bayesian linear regression and linear basis expansions (Section 2.1). We then discuss GP regression (Section 2.2). These topics are then connected via the sparse spectrum and Hilbert space approximations, which utilize linear basis expansions to approximate GPs (Section 2.3). Finally, we discuss BMA (Section 2.4), which can be used to combine distributions from several different models.

### 2.1 Linear Basis Expansions

A common setting for Bayesian linear regression is with an i.i.d. Gaussian prior on a $D$-dimensional weight vector $\boldsymbol{\theta}$ and i.i.d. Gaussian noise, i.e.,

$$\mathbf{y} = \mathbf{X}^\top \boldsymbol{\theta} + \boldsymbol{\varepsilon}, \tag{1}$$

$$\boldsymbol{\theta} \sim \mathcal{N}(\boldsymbol{\theta}; \mathbf{0}, \sigma_{\boldsymbol{\theta}}^2 \boldsymbol{I}_D), \tag{2}$$

$$\boldsymbol{\varepsilon} \sim \mathcal{N}(\boldsymbol{\varepsilon}; \mathbf{0}, \sigma_{\epsilon}^2 \boldsymbol{I}_N), \tag{3}$$

where $\mathbf{X}$ is a $D \times N$ matrix $[\mathbf{x}_1 \cdots \mathbf{x}_N]$ of input data, $\mathbf{y}$ is an $N$-dimensional output vector, $\mathcal{N}$ stands for a Gaussian distribution, $\boldsymbol{\varepsilon}$ is an $N$-dimensional Gaussian vector, $\boldsymbol{I}_D$ and $\boldsymbol{I}_N$ are identity matrices of dimensions $D$ and $N$, respectively, and $\sigma_{\boldsymbol{\theta}}^2$ and $\sigma_{\epsilon}^2$ are variances corresponding to $\boldsymbol{\theta}$ and $\boldsymbol{\varepsilon}$, respectively. If an intercept term is desired, $\mathbf{X}$ can be augmented with a row of ones and $\boldsymbol{\theta}$ taken to be $(D+1)$-dimensional.

By swapping each $\mathbf{x}_n$ with its mapping under a function $\boldsymbol{\phi} \colon \mathbb{R}^D \to \mathbb{R}^F$, the Bayesian linear regression of Eq. (1) becomes a *linear basis expansion*, with Eqs. (2) to (4) being replaced by

$$\mathbf{y} = \boldsymbol{\Phi}^\top \boldsymbol{\theta} + \boldsymbol{\varepsilon}, \tag{4}$$

$$\boldsymbol{\theta} \sim \mathcal{N}(\boldsymbol{\theta}; \mathbf{0}, \sigma_{\boldsymbol{\theta}}^2 \boldsymbol{I}_F), \tag{5}$$

$$\boldsymbol{\varepsilon} \sim \mathcal{N}(\boldsymbol{\varepsilon}; \mathbf{0}, \sigma_{\epsilon}^2 \boldsymbol{I}_N), \tag{6}$$

where $\boldsymbol{\Phi}$ is an $F \times N$ matrix, $[\boldsymbol{\phi}(\mathbf{x}_1) \cdots \boldsymbol{\phi}(\mathbf{x}_N)]$. The functions $\phi_k(\cdot) = \mathrm{proj}_k\big(\phi(\cdot)\big)$ are known as *basis functions*. We refer to $\boldsymbol{\Phi}$ as the "design matrix" of $\mathbf{X}$ with its rows representing the basis functions.

#### 2.1.1 Predictive Distribution and Online Estimation

The model described by Eqs. (2) to (4) is conjugate and Gaussian, which means that efficient, closed-form posterior updates and predictive distributions are available (see, e.g., (Bishop, 2007, Chapter 3)). In particular, let $\boldsymbol{\mu}_{\boldsymbol{\theta}}$ and $\boldsymbol{\Sigma}_{\boldsymbol{\theta}}$ denote the mean and covariance function of the posterior $p(\boldsymbol{\theta}|\mathbf{X}, \mathbf{y}) = \mathcal{N}(\boldsymbol{\theta}|\boldsymbol{\mu}_{\boldsymbol{\theta}}, \boldsymbol{\Sigma}_{\boldsymbol{\theta}})$. Then the posterior predictive distribution of $N_*$ test output/input points, $\mathbf{y}_*$ and $\mathbf{X}_*$, respectively, is given by another Gaussian, i.e.,

$$p(\mathbf{y}_*|\mathbf{y}, \mathbf{X}, \mathbf{X}_*) = \mathcal{N}\big(\mathbf{y}_*|\boldsymbol{\mu}_{y_*}, \boldsymbol{\Sigma}_{y_*}\big), \tag{7}$$

$$\boldsymbol{\mu}_{y_*} = \boldsymbol{\Phi}_*^\top \boldsymbol{\mu}_{\boldsymbol{\theta}}, \tag{8}$$

$$\boldsymbol{\Sigma}_{y_*} = \boldsymbol{\Phi}_*^\top \boldsymbol{\Sigma}_{\boldsymbol{\theta}} \boldsymbol{\Phi}_* + \mathbf{I}_{N_*} \sigma_{\epsilon}^2, \tag{9}$$

where $\boldsymbol{\Phi}_*$ is the design matrix of $\mathbf{X}_*$.

Using the predictive distribution in Eq. (7), online regression can be performed by what is often called "predicting then correcting" in the filtering literature (see, e.g., (Särkkä & Svensson, 2023)). To set up notation, let $\mathbf{x}_{1:t}$ and $y_{1:t}$ denote the first $t$ data points, $\boldsymbol{\Phi}_t$ denote the design matrix of $\mathbf{x}_{1:t}$, and $\boldsymbol{\mu}_{\boldsymbol{\theta},t}, \boldsymbol{\Sigma}_{\boldsymbol{\theta},t}$ denote the parameters of the posterior $p(\boldsymbol{\theta}|\mathbf{x}_{1:t}, y_{1:t}) = \mathcal{N}(\boldsymbol{\theta}|\boldsymbol{\mu}_{\boldsymbol{\theta},t}, \boldsymbol{\Sigma}_{\boldsymbol{\theta},t})$. Then given $\mathbf{x}_{t+1}$ and $y_{t+1}$, the posterior can be updated by the following two steps (Särkkä & Svensson, 2023):

- *(Predict)* Obtain $\mu_{y+1}$ and $\sigma_{y+1}^2$ from Eqs. (8) and (9),

- *(Correct)* Update the posterior to $p(\boldsymbol{\theta}|\mathbf{x}_{1:t+1}, y_{1:t+1}) = \mathcal{N}(\boldsymbol{\theta}|\boldsymbol{\mu}_{\boldsymbol{\theta},t+1}, \boldsymbol{\Sigma}_{\boldsymbol{\theta},t+1})$, where

$$\boldsymbol{\mu}_{\boldsymbol{\theta},t+1} = \boldsymbol{\mu}_{\boldsymbol{\theta},t} + \frac{\boldsymbol{\Sigma}_{\boldsymbol{\theta},t} \phi(\mathbf{x}_{t+1})(y_{t+1} - \mu_{y_{t+1}})}{\sigma_{y_{t+1}}^2}, \tag{10}$$

$$\boldsymbol{\Sigma}_{\boldsymbol{\theta},t+1} = \boldsymbol{\Sigma}_{\boldsymbol{\theta},t} - \frac{\boldsymbol{\Sigma}_{\boldsymbol{\theta},t} \phi(\mathbf{x}_{t+1}) \phi(\mathbf{x}_{t+1})^\top \boldsymbol{\Sigma}_{\boldsymbol{\theta},t}}{\sigma_{y_{t+1}}^2}. \tag{11}$$

### 2.1.2 Fitting Hyperparameters with Empirical Bayes

The approach of Bayesian linear basis expansions is attractive in that it is simple, has closed-form posterior updates, and as we will see, can be quite effective. However, there are several hyperparameters to the model, namely the parameters of the priors $\sigma_{\boldsymbol{\theta}}^2$ and $\sigma_{\epsilon}^2$. It is also typically the case that $\boldsymbol{\phi}(\cdot)$ depends on additional hyperparameters, which we refer to as $\boldsymbol{\psi}$.

A conventional approach to address the lack of knowledge of these parameters is to maximize the marginal likelihood; we can immediately identify this as the prior predictive, and using Eq. (7), calculate it as $p(\mathbf{y}|\mathbf{X}; \sigma_{\boldsymbol{\theta}}^2, \sigma_{\epsilon}^2, \boldsymbol{\psi}) = \mathcal{N}\big(\mathbf{y}|\mathbf{0}, \boldsymbol{\Phi}^\top \boldsymbol{\Phi} \sigma_{\boldsymbol{\theta}}^2 + \boldsymbol{I}_N \sigma_{\epsilon}^2\big)$.[1] This technique is known by several names, including "empirical Bayes," "type-II maximum likelihood estimation," and "evidence maximization" (Theodoridis, 2020, p. 611). Empirical Bayes has a long history of being effectively used for Bayesian linear basis expansions and Gaussian process regression (Rasmussen & Williams, 2005; Bishop, 2007).

Another approach is rooted in the assumption of conjugate priors. For example, closed-form posterior updates are still possible if the joint prior of the linear parameters $\boldsymbol{\theta}$ and the unknown variance $\sigma_{\epsilon}^2$ follows a multivariate normal–inverse Gamma distribution. In this case, it can be demonstrated that the marginal predictive distribution of $y_*$ (i.e., the predictive distribution after integrating out the unknowns $\boldsymbol{\theta}$ and $\sigma_{\epsilon}^2$) is a Student's t-distribution (Zellner, 1996). This analysis has previously been used for a generalizations of IE-GPs in Liu et al. (2023a).

### 2.1.3 Examples of Basis Functions

Many examples of basis functions exist, and we only provide a few here.

The simplest possible basis function is the identity function $\phi(\mathbf{x}) = \mathbf{x}$, in which case we recover ordinary linear regression. Slightly more sophisticated is the polynomial regression, where for scalar $x$, $\phi_{k+1}(x) = x^k$. In either case, an intercept term $a_0$ can be added by appending a row of ones to $\Phi$. Multivariable polynomial regression is formulated similarly with polynomials over $x_1, \ldots, x_D$, but becomes unwieldy with growing dimension, as the number of terms grows exponentially.

From polynomial regression come splines, which are piecewise polynomials between locations $\mathbf{x}_1, \ldots, \mathbf{x}_K$ called knots (de Boor, 1978). From the perspective of a generalized linear model, a design matrix can be calculated for so-called basis splines (B-splines) (de Boor, 1972); however, like polynomials, generalizing splines to high-dimensional settings is expensive and not straightforward. Due to their added complexity, B-splines are not included in the experiments of the current work, but we note they form an interesting topic for research.

Another common example is the set of (Gaussian) radial basis functions (RBFs). They depend on location parameters $\boldsymbol{\mu}_k$ and a length scale parameter $\ell$, i.e.,

$$\phi_k(\mathbf{x}; \boldsymbol{\mu}_k) = \exp\left(-\frac{(\mathbf{x} - \boldsymbol{\mu}_k)^\top (\mathbf{x} - \boldsymbol{\mu}_k)}{2\ell^2}\right).$$

The locations $\boldsymbol{\mu}_k$ may be optimized or taken to be fixed by some initialization (for example, via a K-means clustering of the data).

In what follows, we will introduce GPs and two basis expansions designed to approximate them.

## 2.2 Gaussian Process Regression

Gaussian processes (GPs) are an infinite-dimensional generalization of the linear basis expansion model given above, in the sense that there may be an infinite number of basis functions $\phi_k(\cdot)$ (Rasmussen & Williams, 2005). Inference is then made possible by the *kernel trick* (Rasmussen & Williams, 2005, pp. 12), where basis functions are instead described by a kernel function $\kappa(\cdot, \cdot)$ which summarizes the pairwise relationships of inputs. Indeed, each of the linear basis expansion models explored above is a special case of a Gaussian

---

[1]For numerical stability, when $F < N$, one should use the equivalent expression given in Bishop (2007, pp.167). Regardless of $F$ and $N$, calculations should be handled in the log domain using Cholesky decompositions.

process and can be written with the kernel trick using the notion of the *equivalent kernel* (Bishop, 2007, pp. 159).

While this description of a GP is correct, it is more conventional to describe GPs as a prior distribution over functions. To this end, we must specify a mean function $m(\cdot)$ in addition to the kernel $\kappa(\cdot, \cdot)$; then GP regression (GPR) consists of estimating $f(\cdot)$ in the problem

$$\mathbf{y} = f(\mathbf{x}) + \epsilon, \tag{12}$$

$$f(\mathbf{x}) \sim \mathcal{GP}(m(\mathbf{x}), \kappa(\mathbf{x}, \mathbf{x}')), \tag{13}$$

$$\epsilon \overset{\text{iid}}{\sim} \mathcal{N}(\epsilon; \mathbf{0}, \sigma_\epsilon^2). \tag{14}$$

### 2.2.1 Predictive Distribution

Estimating the value of $f$ on test points in closed form is simple using the conjugate model of Eqs. (12) to (14). As before, let $\mathbf{X}_*$ be a set of test points and $f(\mathbf{X}_*)$ be test values to be inferred. Additionally, let $K(\mathbf{X}_1, \mathbf{X}_2)$ denote the matrix whose entries are given by $[K(\mathbf{X}_1, \mathbf{X}_2)]_{ij} = \kappa(\mathbf{X}_{1\cdot,i}, \mathbf{X}_{2\cdot,j})$. Then the predictive distribution is again given by a Gaussian distribution,

$$p(f(\mathbf{X}_*)|\mathbf{y}, \mathbf{X}, \mathbf{X}_*) = \mathcal{N}(f(\mathbf{X}_*)|\boldsymbol{\mu}_{f_*}, \boldsymbol{\Sigma}_{f_*}), \tag{15}$$

$$\boldsymbol{\mu}_{f_*} = m(\mathbf{X}_*) + K(\mathbf{X}_*, \mathbf{X})\big[K(\mathbf{X}, \mathbf{X}) + \sigma_\epsilon^2 \boldsymbol{I}\big]^{-1}(\mathbf{y} - m(\mathbf{X})), \tag{16}$$

$$\boldsymbol{\Sigma}_{f_*} = K(\mathbf{X}_*, \mathbf{X}_*) - K(\mathbf{X}_*, \mathbf{X})\big[K(\mathbf{X}, \mathbf{X}) + \sigma_\epsilon^2 \boldsymbol{I}\big]^{-1} K(\mathbf{X}, \mathbf{X}_*). \tag{17}$$

In contrast to the basis expansion case, the values $f(\mathbf{X}_*)$ are noiseless evaluations of the random function $f$. From Eq. (12), it is easy to see that the predictive distribution of the noisy quantity $\mathbf{y}_*$ can be obtained by simply adding $\boldsymbol{I}_{N_*}\sigma_\varepsilon^2$ to the predictive covariance in Eq. (15).

These expressions are simple, but Eqs. (16) and (17) require solving the linear systems

$$\big[K(\mathbf{X}, \mathbf{X}) + \sigma_\epsilon^2 \boldsymbol{I}\big]\boldsymbol{b} = (\mathbf{y} - m(\mathbf{X})) \quad \text{and} \quad \big[K(\mathbf{X}, \mathbf{X}) + \sigma_\epsilon^2 \boldsymbol{I}\big]\mathbf{c} = K(\mathbf{X}, \mathbf{X}_*).$$

By using the Cholesky decomposition of $\big[K(\mathbf{X}, \mathbf{X}) + \sigma_\epsilon^2 \boldsymbol{I}\big]$ and solving the associated triangular systems, the calculation of Eqs. (16) and (17) can be made numerically stable and memory efficient relative to the calculation of inverses, but incurs $\mathcal{O}(N^3)$ time complexity (Rasmussen & Williams, 2005, App. A). The poor computational scaling with respect to $N$ of solving these linear systems is a major obstacle to implementing GPs in practice and will motivate our discussion of scalable approximations. But first, we will specify the mean and covariance functions that are most commonly used.

### 2.2.2 The Choice of Mean and Kernel

Without loss of generality, the mean function $m(\cdot)$ can be chosen to be identically zero. The kernel function $\kappa(\cdot, \cdot)$ can in principle be any positive semi-definite function, but a few choices of kernel dominate the literature. The squared exponential (SE) kernel and the Matérn-3/2 kernels are likely the most common (Rasmussen & Williams, 2005, Chapter 4):

$$\kappa_{\text{SE}}(\mathbf{x}, \mathbf{x}') = \sigma_f^2 \exp\left(\frac{-\|\mathbf{x} - \mathbf{x}'\|^2}{2\ell^2}\right),$$

$$\kappa_{\text{Matérn-3/2}}(\mathbf{x}, \mathbf{x}') = \sigma_f^2\left(1 + \frac{\sqrt{3}\|\mathbf{x} - \mathbf{x}'\|}{\ell}\right) \exp\left(\frac{-\sqrt{3}\|\mathbf{x} - \mathbf{x}'\|}{\ell}\right),$$

where $\ell$ and $\sigma_f$ are the length scale and process scale hyperparameters, and are both positive. The process scale $\sigma_f$ is directly related to prior variance $\sigma_{\boldsymbol{\theta}}^2$ of the linear basis expansion case, and in subsequent sections, we will refer to $\sigma_f$ as $\sigma_{\boldsymbol{\theta}}$ when convenient.

We can additionally allow for the length scale to vary in each dimension; this modification to kernels is called *automatic relevance determination (ARD)*. By expressing

$$r_{\text{ARD}}^2(\mathbf{x}_1, \mathbf{x}_2) = \sum_{d=1}^{D} \frac{(x_{1,d} - x_{2,d})^2}{\ell_d^2} = \|\mathbf{x}_1 - \mathbf{x}_2\|_{\text{diag}(\ell_1, \dots, \ell_D)}^2,$$

the SE-ARD and Matérn-3/2-ARD kernels can be expressed as follows:

$$\kappa_{\text{SE-ARD}}(\mathbf{x}, \mathbf{x}') = \sigma_f^2 \exp\left(-0.5 r_{\text{ARD}}^2\right),$$
$$\kappa_{\text{Matérn-3/2-ARD}}(\mathbf{x}, \mathbf{x}') = \sigma_f^2 \left(1 + \sqrt{3} r_{\text{ARD}}\right) \exp\left(-\sqrt{3} r_{\text{ARD}}\right).$$

To fully specify a kernel function, optimal values of $\sigma_f$ and $\ell$ must also be determined. In this regard, it is common to employ the empirical Bayes approach discussed in Section 2.1.2. Again similar to the linear basis expansion case, the marginal likelihood is easily seen to be $\mathcal{N}(\mathbf{y}|\mathbf{0}, K(\mathbf{X}, \mathbf{X}) + \sigma_\epsilon^2 \mathbf{I})$.

## 2.3 Sparse Spectrum and Hilbert Space Approximations

While GPs are a flexible tool that makes minimal assumptions on the data-generating process, calculating the predictive distribution requires the (symbolic) inversion of an $N \times N$ matrix, or alternatively, the solving of the corresponding linear system. This is a $\mathcal{O}(N^3)$ operation, and so quickly becomes unwieldy as $N$ grows. A typical way to deal with this is the inclusion of *inducing points*, a set of $M < N$ points that are chosen to summarize the data (Quinonero-Candela & Rasmussen, 2005). An alternative approach is to take a finite basis expansion via the spectra of the GPs, as is done in the random Fourier feature (RFF) (Lázaro-Gredilla et al., 2010) and Hilbert space GP (HSGP) (Solin & Särkkä, 2020) approximations.

### 2.3.1 Random Fourier Feature Gaussian Processes

The main idea of RFF GPs is to take a Monte Carlo approximation of the Wiener-Khinchin integral, which we describe below. The kernel of a GP is said to be *stationary* if it depends only on the difference $\mathbf{x} - \mathbf{x}'$. In this case, Bochner's theorem guarantees a spectral representation of the kernel in the form of the Wiener-Khinchin integral (Rasmussen & Williams, 2005, pp. 82):

$$\kappa(\mathbf{x}, \mathbf{x}') = \int_{\mathbb{R}^D} \exp(2\pi i \mathbf{s}^\top (\mathbf{x} - \mathbf{x}')) \, d\mu(\mathbf{s}) \tag{18}$$

for some positive finite measure $\mu(\cdot)$.

In the case that $\mu(\cdot)$ admits a density $S(\mathbf{s})$ w.r.t. the Lebesgue measure on $\mathbb{R}^D$, it is known as the *power spectral density (PSD)* and the Wiener-Khinchin theorem says it is given by the Fourier transform of $g(\mathbf{x} - \mathbf{x}') = \kappa(\mathbf{x}, \mathbf{x}')$ (Rasmussen & Williams, 2005, pp. 82). Then we can sample $\mathbf{s}_1, \ldots, \mathbf{s}_F \sim S(\mathbf{s})$ and approximate Eq. (18) by

$$\kappa(\mathbf{x}, \mathbf{x}') \approx \sum_{m=1}^{F} \exp(2\pi i \mathbf{s}_m^\top (\mathbf{x} - \mathbf{x}')). \tag{19}$$

By noting the symmetry of the PSD, and to ensure real-valued kernels, we in practice sample $F/2$ spectral points $\mathbf{s}_1, \ldots, \mathbf{s}_{F/2}$ and use both $\mathbf{s}_m$ and $-\mathbf{s}_m$ in Eq. (19) (Lázaro-Gredilla et al., 2010). This results in a linear basis expansion model a la Eq. (4) with the basis function

$$\phi(\mathbf{x}) = \sqrt{\frac{2}{F}} [\sin(\mathbf{x}^\top \mathbf{s}_1) \, \cos(\mathbf{x}^\top \mathbf{s}_1) \, \cdots \, \sin(\mathbf{x}^\top \mathbf{s}_{F/2}) \, \cos(\mathbf{x}^\top \mathbf{s}_{F/2})]^\top \in \mathbb{R}^F. \tag{20}$$

### 2.3.2 Hilbert Space Gaussian Processes

Hilbert space GPs (Solin & Särkkä, 2020) take a different approach, which can be viewed as a quadrature rule on the same integral of Eq. (18). Their derivation is rooted in targeting the power spectral density and approximating it via Hilbert space methods from partial differential equations and can be found in Solin & Särkkä (2020). We begin by specifying $\phi(\cdot)$ in the scalar case, $D = 1$.

For the Hilbert space approximation to be taken, a compact subset of $\Omega \subset \mathbb{R}$ must be specified where the approximation is valid. Riutort-Mayol et al. (2023) explore the case where $\Omega = [-L, L]$, with $L = c \max_{\mathcal{D}} |x|$, representing the maximum absolute value $x$ takes on over the training data $\mathcal{D}$ scaled by a value $c > 1$.

The resulting approximation is given by the basis functions (Solin & Särkkä, 2020)

$$\phi_k(x) = S\left(\frac{k\pi}{2L}\right)\frac{\sin\left(\frac{k\pi}{2L}(x+L)\right)}{\sqrt{L}}, \quad k = 1, \ldots, F, \tag{21}$$

where we recall $S(s)$ to be the PSD of the kernel.

The multivariable case is written similarly (Solin & Särkkä, 2020). However, the approximation suffers from exponential growth, in the sense that using $F_1, \ldots, F_D$ basis functions for each input dimension results in a $F_1 \cdot F_2 \cdots F_D$ dimensional embedding. As a result, the approximation is typically more efficient than the exact GP only when $D \lesssim 3$ (Riutort-Mayol et al., 2023).

In low dimensions ($D \lesssim 3$), the HSGPs empirically converge to the true GP much more quickly than RFF GPs (Solin & Särkkä, 2020; Riutort-Mayol et al., 2023). While HSGPs are not tractable in high dimensions, we note that RFF GPs also suffer harshly from the curse of dimensionality, which is typical in Monte Carlo sampling. In Section 4.3, we argue why better performance in the univariate case is sufficiently attractive.

### 2.4 Bayesian Model Averaging

Bayesian model averaging is a long-standing technique to combine Bayesian models (Hoeting et al., 1999). With a sequence of models $\mathcal{M}_1, \ldots, \mathcal{M}_M$, BMA computes the distribution $p(y|\mathcal{D})$ by a process that simulates the marginalization of the choice of model, i.e.,

$$p(y|\mathcal{D}) \propto \sum_{m=1}^{M} \Pr(\mathcal{M}_m|\mathcal{D})p(y|\mathcal{D}, \mathcal{M}_m). \tag{22}$$

The resulting distribution can be understood as a mixture model with mixands $p(y|\mathcal{D}, \mathcal{M}_m)$ and weights $w_m = \Pr(\mathcal{M}_m|\mathcal{D})$.

BMA is simple and asymptotically optimal when the true data-generating process is one of the models $\mathcal{M}_1, \ldots, \mathcal{M}_M$ (Hoeting et al., 1999). This is on account of "collapsing" in the limit of infinite data, in the sense that $\Pr(\mathcal{M}_m|\mathcal{D})$ approaches 1 for the $\mathcal{M}_m$, which minimizes a statistical distance to the true data-generating distribution (Yao et al., 2018).

## 3 Dynamic Online Ensembles of Basis Expansions

In this section. we re-introduce the work of Lu et al. (2022) with no explicit reference to GPs. This comprises an online learning and ensembling step (Section 3.1), as well as an adjustment for problems where dynamics (Section 3.2) or switching (Section 3.3) are present. Finally, we discuss adaptations to non-Gaussian likelihoods (Section 3.4) and theoretical regret analysis (Section 3.5).

### 3.1 Online Ensembles of Basis Expansions

The Online Ensemble of Basis Expansions (OEBE) consists of learners with basis expansions $\phi^{(m)}(\cdot)$ and parameters $\boldsymbol{\theta}^{(m)}$ for $m = 1, \ldots, M$, and a meta-learner based on BMA. The meta-learner initializes weights $\boldsymbol{w}_t$ to a uniform prior over models, and then learning begins.

After the retrieval of a new data pair $(\mathbf{x}_{t+1}, y_{t+1})$, each learner predicts $p(y_{t+1}|\mathbf{x}_{1:t}, \mathbf{y}_{1:t}, \mathbf{x}_{t+1})$ via Eqs. (7) to (9). The BMA weights are updated as follows:

$$w_{t+1}^{(m)} \propto w_t^{(m)} p^{(m)}(y_{t+1}|\mathbf{x}_{1:t}, \mathbf{y}_{1:t}, \mathbf{x}_{t+1}). \tag{23}$$

Note that, if a weight $w_t^{(m)}$ (numerically) collapses to 0 at some time $t$, it will remain at 0.

Next, posteriors $p(\boldsymbol{\theta}_{t+1}^{(m)}|\mathbf{x}_{1:t+1}, \mathbf{y}_{1:t+1})$ are calculated via Eqs. (10) and (11). Finally, we note that the minimum mean squared error (MMSE) estimate[2] of $y_{t+1}$ is given by

$$\mu_{y_{t+1}} = \sum_m w_t^{(m)} \mu_{y_{t+1}}^{(m)}, \tag{24}$$

and the variance of this estimate by

$$\sigma_{y_{t+1}}^2 = \sum_m w_t^{(m)} \left( \sigma_{y_{t+1}}^{(m)}{}^2 + \left( \mu_{y_{t+1}} - \mu_{y_{t+1}}^{(m)} \right)^2 \right). \tag{25}$$

The OEBE contains as a special case the IE-GP if $\phi^{(m)}(\cdot)$ is given by Eq. (20) for every $m$. However, the functions $\phi^{(m)}(\cdot)$ can be of any type of basis expansion, including different types of basis expansions for different values of $m$.

Note that while each model is linear in $\boldsymbol{\theta}^{(m)}$, each $\boldsymbol{\theta}^{(m)}$ corresponds to a different (nonlinear) basis expansion $\phi^{(m)}(\cdot)$. Therefore each $\boldsymbol{\theta}^{(m)}$ represents a different quantity, perhaps even defined on a different space. In particular, even with the MMSE estimate Eq. (24), the resulting estimate is a mixture of several *different* nonlinear functions of $\mathbf{x}_t$.

We remark that with the MMSE estimate, the resulting prediction is linear in the "stacked" vector $\boldsymbol{\theta} = [\boldsymbol{\theta}^{(1)} \cdots \boldsymbol{\theta}^{(m)}]$, but this does not reflect how each model is trained. For example, an ensemble consisting of the basis expansions

$$\phi^{(1)}(\mathbf{x}) = \begin{bmatrix} 1 \\ x \end{bmatrix} \quad \text{and} \quad \phi^{(2)}(\mathbf{x}) = \begin{bmatrix} x^2 \end{bmatrix}$$

is not equivalent to the quadratic basis expansion model. Depending on the basis expansions used, this may result in a less expressive model (as in the example above), but it also lessens the computational burden of empirical Bayes significantly, which scales cubically with $F$.

## 3.2 Adding a Random Walk on the Parameters

If the dynamics of a problem are expected to change in the dataset, we can add a random walk on the parameters $\boldsymbol{\theta}$. For the RFF GP case, this was referred to as a *dynamic IE-GP (DIE-GP)*, so we follow in calling the generalization a *dynamic OEBE (DOEBE)*. To differentiate OEBE from DOEBE, we will often refer to OEBE models as "static," juxtaposed to "dynamic."

Incorporating dynamics can be done simply by injecting a noise term $\boldsymbol{\varepsilon}_{\text{rw},t}^{(m)}$ with variance $\sigma_{\text{rw}}^{(m)}{}^2$ on the posterior $\boldsymbol{\theta}^{(m)}$ prior to prediction.

$$p(\boldsymbol{\theta}_{t+1}^{(m)}|\mathbf{X}_{1:t}, \mathbf{y}_{1:t}) = \mathcal{N}(\boldsymbol{\theta}_{t+1}^{(m)}|\boldsymbol{\mu}_{\boldsymbol{\theta},t}^{(m)}, \boldsymbol{\Sigma}_{\boldsymbol{\theta},t}^{(m)} + \sigma_{\text{rw}}^{(m)}{}^2 \boldsymbol{I}). \tag{26}$$

Note that this slightly differs from the formulation in Lu et al. (2022), which does not allow $\sigma_{\text{rw}}^{(m)}{}^2$ to vary with $m$. We somewhat generalize to allow OEBE and DOEBE models to be ensembled together, as in Section 4.2. Additionally, note that OEBE is the special case where noise is degenerate, i.e., when $\sigma_{\text{rw}}^{(m)} = 0$.

From a signal processing perspective, each model in the DOEBE can be viewed as the Kalman filter formulation for a linear basis expansion with drift, with Eq. (26) being a closed form solution of the Chapman-Kolmogorov equation (Särkkä & Svensson, 2023, pp. 34). Accordingly, the $m$th learner corresponds to a state space model

$$\boldsymbol{\theta}_{t+1}^{(m)} = \boldsymbol{\theta}_t^{(m)} + \boldsymbol{\varepsilon}_{\text{rw},t}^{(m)}$$
$$y_t^{(m)} = \boldsymbol{\phi}^{(m)}(\mathbf{x}_t)^\top \boldsymbol{\theta}_t^{(m)} + \varepsilon_t$$

---

[2]In principle, we could record the entire Gaussian mixture model as the estimate. Because of the rapid collapse of BMA to a single mixand, it does not much change the results, and simplifies computations.

$$\tilde{w}_{t+1} = \boldsymbol{Q}\boldsymbol{w}_t \qquad\qquad\qquad \text{SDOEBE}$$

$$\boldsymbol{\theta}_{t+1}^{(m)} = \boldsymbol{\theta}_t^{(m)} + \boldsymbol{\varepsilon}_{\text{rw},t}^{(m)} \qquad\qquad\qquad \text{DOEBE}$$

$$y_t^{(m)} = \boldsymbol{\phi}^{(m)}(\mathbf{x}_t)^\top \boldsymbol{\theta}_t^{(m)} + \varepsilon_t$$
$$p(y_{t+1}|\mathbf{X}_{1:t}, \mathbf{y}_{1:t}, \mathbf{x}_{t+1}) = \sum_m w_t^{(m)} p^{(m)}(y_{t+1}|\mathbf{X}_{1:t}, \mathbf{y}_{1:t}, \mathbf{x}_{t+1})$$
$$w_{t+1}^{(m)} \propto \tilde{w}_{t+1}^{(m)} p^{(m)}(y_{t+1}|\mathbf{X}_{1:t}, \mathbf{y}_{1:t}, \mathbf{x}_{t+1}) \qquad\qquad \text{OEBE}$$

Figure 1: The hierarchy of the proposed models. All models have the same distribution for $y_t$ conditioned on $\boldsymbol{\theta}_t$, while DOEBE adds a random walk to parameters, and SDOEBE adds a random walk to BMA weights.

where $\varepsilon_{\text{rw},t}^{(m)} \overset{\text{iid}}{\sim} \mathcal{N}(\varepsilon_{\text{rw},t}^{(m)}; \mathbf{0}, \sigma_{\text{rw}}^{(m)2}\boldsymbol{I})$. The state-space formulation shows how this random walk may be even more complex, i.e.,

$$\boldsymbol{\theta}_{t+1}^{(m)} = \boldsymbol{\theta}_t^{(m)} + \boldsymbol{L}^{1/2}\boldsymbol{\varepsilon}_t,$$

where $\boldsymbol{L}^{1/2}$ is the Cholesky factor of a positive semi-definite matrix and $\boldsymbol{\varepsilon}_t$ is white Gaussian noise. While this more general form may be useful in some applications, particularly when domain expertise is available, we choose to focus on the special case $\boldsymbol{L} = \sigma_{\text{rw}}^{(m)2}\boldsymbol{I}$ in the current work for simplicity.

From this state-space perspective, we can also explore a formal justification for placing a random walk on the parameters. Namely, if the estimation task is expected to change slightly over time, it is useful to model this change as a drift in the model parameters. From a statistical perspective, the posterior distribution of Eq. (26) merely appears as a "more conservative" estimate of the posterior, but this is beneficial in the recursive setting: since we use the posterior at time $t$ as the prior for time $t+1$, a more conservative posterior at time $t$ corresponds exactly to a weaker prior at time $t+1$. This allows for better adaptation to new data by "forgetting" old data.

As the weight update step remains unchanged, the DOEBE inherits the weight collapse property of the OEBE. Likewise, the DOEBE contains the DIE-GP as a special case when the noise hyperparameters $\sigma_{\text{rw}}^{(m)}$ are all the same and every basis function $\boldsymbol{\phi}^{(m)}(\cdot)$ is given by Eq. (20).

### 3.3 Adding a Random Walk to the BMA Weights

Allowing for a random walk on the BMA weights is also a simple addition, which becomes important for ensembling dynamic and static models. Analogous to the Gaussian random walk on $\boldsymbol{\theta}$, dynamics can be introduced to $\boldsymbol{w}_t$ via a discrete-time Markov chain, specified by a stochastic matrix $\boldsymbol{Q}$:

$$\boldsymbol{w}_{t+1} = \boldsymbol{Q}\boldsymbol{w}_t \iff w_{t+1}^{(m)} = \sum_{m'=1}^{M} Q_{mm'} w_t^{(m')}. \tag{27}$$

We follow Lu et al. (2022) in calling this a *switching (dynamic) OEBE (S(D)OEBE)*.

The term "switching" is suggestive of applications where regime switching (i.e., sudden changes from one generating model to another) may be present. Additionally, Lu et al. (2022) show that adding switching allows for regret analysis in adversarial settings.

However, the more important property introduced by Eq. (27) to the current work is that the revival of collapsed weights is possible (Lu et al., 2022). In particular, note that even if $w_t^{(m)}$ has (numerically) collapsed to 0, it can be revived so long as $Q_{mm'} w_t^{(m')}$ is positive for some $m'$. This allows us to preserve the full ensemble structure after the numerical collapse of the BMA weights, which often occurs in the first few thousand samples.

The OEBE, DOEBE, and SDOEBE models, as well as the relationships between them, are summarized in Fig. 1.

### 3.4 Inference With Non-Gaussian Likelihoods

One seeming weakness of the DOEBE approach is the necessity of Gaussian likelihoods. However, by casting the DOEBE as a linear Gaussian state-space model, we are able to use several approximations for arbitrary non-Gaussian likelihoods.

The simplest such approximation is the Laplace approximation (Kass et al., 1991), which is used for IE-GPs. The Laplace approximation takes a second-order Taylor series approximation of the log-posterior $\log p(\boldsymbol{\theta}_t^{(m)}|\mathbf{X}_{1:t}, \mathbf{y}_{1:t})$ centered around the MAP estimate of $\boldsymbol{\theta}$, which in turn is the solution to the optimization problem

$$\boldsymbol{\theta}_{t;\text{MAP}}^{(m)} = \arg\max_{\boldsymbol{\theta}} p(\boldsymbol{\theta}_t^{(m)}|\mathbf{X}_{1:t}, \mathbf{y}_{1:t}). \tag{28}$$

The resulting approximation is a Gaussian distribution whose covariance is given by the inverse of the Hessian matrix for $-\log p(\boldsymbol{\theta}_t^{(m)}|\mathbf{X}_{1:t}, \mathbf{y}_{1:t})$, i.e.,

$$\left(\boldsymbol{\Sigma}_{\boldsymbol{\theta},t;\text{Laplace}}^{(m)}\right)^{-1} = -\nabla_{\boldsymbol{\theta}^{(m)}}^2 \log p(\boldsymbol{\theta}^{(m)}|\mathbf{X}_{1:t}, \mathbf{y}_{1:t})\big|_{\boldsymbol{\theta}^{(m)}=\boldsymbol{\theta}_{t;\text{MAP}}^{(m)}}. \tag{29}$$

By noting the linearity of the Laplacian and factorizing the posterior $\log p(\boldsymbol{\theta}_t^{(m)}|\mathbf{X}_{1:t}, \mathbf{y}_{1:t})$, the computation of Eq. (29) can be done recursively using only the Hessian with respect to the likelihood, i.e.,

$$\left(\boldsymbol{\Sigma}_{\boldsymbol{\theta},t;\text{Laplace}}^{(m)}\right)^{-1} = \left(\boldsymbol{\Sigma}_{\boldsymbol{\theta},t-1;\text{Laplace}}^{(m)}\right)^{-1} - \nabla_{\boldsymbol{\theta}^{(m)}}^2 \log p(y_t|\boldsymbol{\theta}^{(m)}, \mathbf{x}_t)\big|_{\boldsymbol{\theta}^{(m)}=\boldsymbol{\theta}_{t;\text{MAP}}^{(m)}}. \tag{30}$$

In this work, our principal concern is (Gaussian) regression, so we do not undertake extensive experiments for classification. Nevertheless, we provide a simple experiment using the Laplace approximation in Appendix A where the proposed method is tested on a toy dataset. We show performance comparable to other recent online GP approximations, and demonstrate that dynamic models can outperform static models when the distribution of data changes gradually over time.

If classification is of particular concern, several other approximation methods are available that one could experiment with. For example, particle filtering (Djurić et al., 2003) or posterior linearization filters (Tronarp et al., 2018) offer other methods for inference in state-space models with non-Gaussian measurement models. We believe the adaptation of DOEBE to these non-Gaussian settings is an interesting avenue for future work, and speculate that they may provide better results than the Laplace approximation.

### 3.5 Theoretical Aspects

In Lu et al. (2022), several online regret bounds were derived. The first regret bound is similar to that of Kakade & Ng (2004) and bounds the cumulative loss between the IE-GP and any individual expert of the ensemble. This result does not directly depend on the use of RFF GPs in particular and is easily adapted to the OEBE, with a minor additional assumption on each $\phi^{(m)}$.

For this, we will define the loss function $\mathcal{L}(f(\mathbf{x}_\tau); y_\tau)$ as the negative log-likelihood of data $y_\tau$ using estimator $f(\mathbf{x}_\tau)$. We will define the loss of expert $m$ at time $t+1$, $l_{t+1}^{(m)}$, as the negative predictive log-likelihood of $y_{t+1}$ under the posterior predictive of estimator $m$ at time $t$. Likewise, $\ell_{t+1}$ is the negative predictive log-likelihood of $y_{t+1}$ under the posterior predictive of the OEBE estimator at time $t$. We are then able to bound the gap between the cumulative loss of any expert $m$ with fixed parameters $\boldsymbol{\theta}_*^{(m)}$ and the OEBE estimator.

**Theorem 1** (Online Regret Bound for OEBE). *Let the negative log-likelihood $\mathcal{L}(\cdot; y_\tau)$ be $\mathcal{C}^2$ with bounded second derivative, i.e., $|\frac{d^2}{dz_\tau}\mathcal{L}(z_\tau; y_\tau)| \leq c$ for all $z_\tau$ and some constant $c$. Let us consider an OEBE with prior*

$$p(\boldsymbol{\theta}^{(m)}) = \mathcal{N}\left(\boldsymbol{\theta}^{(m)}; \mathbf{0}, \sigma_{\boldsymbol{\theta}}^{(m)2} \boldsymbol{I}_{F_M}\right),$$

*where $F_m = \dim \boldsymbol{\theta}^{(m)}$. Furthermore, assume that for any $\mathbf{x}$, the norm of $\phi^{(m)}(\mathbf{x})$ is bounded by unity.*

*Then the cumulative negative log-likelihood incurred by the OEBE is bounded with respect to any single expert $m$ with fixed parameters $\boldsymbol{\theta}_*^{(m)}$ by*

$$\sum_{\tau=1}^{T} \ell_t - \sum_{\tau=1}^{T} \mathcal{L}(\boldsymbol{\phi}^{(m)}(\mathbf{x}_\tau)^\top \boldsymbol{\theta}_*^{(m)}; y_\tau) \le \frac{\|\boldsymbol{\theta}_*^{(m)}\|^2}{2\sigma_{\boldsymbol{\theta}}^2} + \frac{F_m}{2} \log\left(1 + \frac{Tc\sigma_{\boldsymbol{\theta}^{(m)}}^2}{F_m}\right) + \log M. \tag{31}$$

*Proof.* The proof follows immediately from the proof of Lemma 1 in Lu et al. (2022, Sec. 9.1), which makes no reference to RFF GPs besides their being a (bounded) basis expansion. For notational consistency and completeness, we provide a proof in Appendix B. $\qquad\square$

The authors of Lu et al. (2022) then show regret bounds with respect to any learner in the RKHS of any expert by essentially using the uniform convergence of RFF GPs (Rahimi & Recht, 2007). A higher probability bound could also be derived with the classical results of Sutherland & Schneider (2015). Similar uniform convergence results exist for HSGPs (Solin & Särkkä, 2020), which we posit can be used to derive similar regret bounds with respect to the RKHS of each expert. However, we do not discuss them here, as our main focus is on the ensembling of several different types of basis expansions.

The switching regret bounds in Lemma 2 and Lemma 3 of Lu et al. (2022) also do not depend on the RFF approximation and can be adapted to SOEBE. The resulting bounds require more bookkeeping on account of the possibly varying dimension of each $\boldsymbol{\theta}^{(m)}$, but the proofs again remain essentially the same as for IE-GPs. We also omit derivation of these bounds to not distract from our main interest, which is more practical in nature.

## 4 Choosing Models for Ensembling

In this section, we make general comments on how an ensemble can be constructed. This includes how to sample kernel hyperparameters (Section 4.1), how to handle problems where it is unclear if a change in the prediction problem will occur (Section 4.2), and why one may want to include generalized additive models in the ensemble (Section 4.3).

### 4.1 Sampling Hyperparameters from the Marginal Likelihood

For IE-GPs, some initial data $\mathcal{D}_0$ is set aside to determine $\sigma_{\boldsymbol{\theta}}^2$ and $\sigma_\epsilon^2$ via empirical Bayes (Section 2.1.2). There is no guidance given on how to obtain kernel hyperparameters. To allow for expressive basis functions depending on many hyperparameters, we allow all hyperparameters to be optimized with respect to the marginal likelihood.

Clearly, there is no diversity among the same type of model in the ensemble if we simply take the estimates via empirical Bayes. We provide a two-pronged approach to promote ensemble diversity, targeting both the multimodality and probabilistic nature of the marginal likelihood.

The first strategy hinges on the observation that marginal likelihoods are often multimodal (Rasmussen & Williams, 2005, pp. 115). To leverage this fact, we optimize with several initial sets of hyperparameters. In practice, we find that initializing models with length scales to $\ell_d = s(\max_{\mathcal{D}_0} x_d - \min_{\mathcal{D}_0} x_d)$ for $s \in \{0.1, 1, 10\}$ works well.[3]

Second, we note that the marginal likelihood is indeed a valid probability distribution and propose taking a Laplace approximation of the marginal likelihood at each local maximum. In particular, we sample $M-1$ sets of hyperparameters from the Laplace approximation, in addition to keeping the empirical Bayes estimate, for a total of $M$ estimates at each mode.[4] This maintains the same spirit of using empirical Bayes for the variance terms whilst allowing for a legitimate ensemble. Additionally, if the Hessian can be computed quickly (e.g., using automatic differentiation when the number of hyperparameters is small), this sampling

---

[3]The $s = 1$ case is used as an initialization in the code of Lázaro-Gredilla et al. (2010).

[4]For optimization, each constrained hyperparameter is transformed to an unconstrained one. The sampling occurs in the unconstrained space.

step is also quick to compute. We observe that this provides better accuracy, significant at the $p < 0.05$ level according to a Wilcoxon signed-rank test (Appendix C).

It is tempting to retrain the hyperparameters periodically as an additional method of dealing with a changing task. While this may be worthwhile in some applications, it has the unfortunate consequence of changing the computational complexity of estimation over time. In particular, if the ensemble is retrained periodically, the online algorithm will drastically increase its complexity periodically, which is often undesirable in an online algorithm. Continual learning of hyperparameters, for example with online stochastic gradient descent also adds unfortunate complexities by changing the measurement model of the state-space without changing the current posterior. In our experiments, we therefore fix all hyperparameters after the initialization above is performed.

## 4.2 Ensembles of Ensembles for Dynamic and Static Models

Dynamic models are expected to perform significantly better than static models if a progressive change in the online estimation problem occurs. However, if the data consists of i.i.d. draws from a GP, it is clear that dynamic models will overestimate the predictive variance (see Eq. (26)). Therefore, it is desirable to be able to ensemble static and dynamic models.

A straightforward solution is to use a DOEBE model where $\sigma_{\mathrm{rw}}^{(m)2}$ is zero for some models and nonzero for others. However, recall from Section 2.4 that a central property of BMA is that it collapses in the limit of infinite data to the model "closest" to the generating model. In practice, this collapse happens rapidly, often within a few thousand samples in our experiments, and weights may become exactly zero due to numerical underflow or implementation. This is highly problematic, as this collapse can (numerically) occur before it becomes clear that dynamic models are necessary.

To theoretically avoid the collapse of BMA weights, any matrix $\boldsymbol{Q}$ which describes an ergodic Markov chain is sufficient.[5] However, the collapse of BMA weights to zero is a numerical property rather than a theoretical property in the first place. Moreover, we can incorporate the intuition that the methods that perform best statically are likely to also perform best dynamically, and vice versa.

We propose an SDOEBE with a particular form of $\boldsymbol{Q}$ which incorporates both these points. In particular, we specify an ensemble of $2M$ models, the first $M$ dynamic and the second $M$ static, and we build an SDOEBE matrix $\boldsymbol{Q}$ with the following block structure:

$$\boldsymbol{Q} = \begin{bmatrix} (1-\delta)\boldsymbol{I}_M & \delta\boldsymbol{I}_M \\ \delta\boldsymbol{I}_M & (1-\delta)\boldsymbol{I}_M \end{bmatrix}.$$

Because of the block structure of $\boldsymbol{Q}$, we call this model an *Ensemble of DOEBEs (E-DOEBEs)*. It can be interpreted as switching between dynamic and static models, where the probability of switching is given by $\delta$.

The E-DOEBE unfortunately introduces the hyperparameter $\delta$, which needs to be selected, in addition to the values of $\sigma_{\mathrm{rw}}^{(m)}$ for each $m$. These hyperparameters are not easily amendable to the empirical Bayes approach either, since their entire purpose is to improve performance on timescales larger than the pretraining period.

In the case of $\delta$, we performed an experiment in Appendix D where the value of $\delta$ is varied from 0.01 to 0.2; we found that the effects are relatively small, and a value of $\delta \simeq 0.05$ works well on all tested datasets. Meanwhile, for $\sigma_{\mathrm{rw}}^2$, the block structure of the E-DOEBE can be repeated to ensemble several values of $\sigma_{\mathrm{rw}}^2$. For example, to ensemble $R$ different models with $\sigma_{\mathrm{rw}}^2 \in \{\sigma_{\mathrm{rw}}^{(1)}, \ldots, \sigma_{\mathrm{rw}}^{(R)}\}$, we would use the $RM \times RM$ matrix with $(1 - R\delta)\boldsymbol{I}_M$ on the block-diagonals and $\delta\boldsymbol{I}_M$ elsewhere.

While the E-DOEBE is a special case of the SDOEBE, it is a novel formulation and consistently outperforms any given DOEBE model in our experiments.

---

[5]This follows immediately from the definition of ergodicity and that the Gaussian distribution's support is all of $\mathbb{R}$.

### 4.3 Why Additive Models May be Preferable

In the case of either RFF GPs or HSGPs, performance degrades rapidly for fixed feature dimension $F$ and increasing input dimension $D$. However, GPs with "additive" structure have been widely explored (Duvenaud et al., 2011). The simplest case is when the GP is given the structure of a generalized additive model (GAM) (Hastie & Tibshirani, 1990), known as a GAM-GP:

$$f(\mathbf{x}) \sim \mathcal{GP}\left(0, \sum_{d=1}^{D} \kappa_d(x_d, x_d')\right) \iff f(\mathbf{x}) = \sum_{d=1}^{D} f_d(x_d), \tag{32}$$

where $f_d(x_d) \sim \mathcal{GP}\left(0, \kappa(x_d, x_d')\right)$ are GPs with respective kernel function $\kappa_d(\cdot, \cdot)$.

In this case, the number of basis functions in the HSGP approximation grows linearly with $D$, and likewise, so does the volume integrated over in the RFF approximation. This can be seen from the right-hand side of Eq. (32), as we can simply take $D$ independent approximations. Therefore, while the GAM-GP structure may be restrictive, we can obtain a better-quality approximation as the input dimension of the data $D$ grows for a fixed number of features $F$. This is useful because, in medium-to-high dimensions[6], the approximation error stemming from high-dimensional quadrature may dominate the approximation error from assuming additive structure.

For large datasets, Hensman et al. (2018) show how a variational version of RFF GPs with additive structure can be effective and efficient, and Solin & Särkkä (2020) show the same for HSGPs. Delbridge et al. (2020) explore GAM-GPs and a generalization using random projections and note their "surprisingly" good performance on large datasets and in high dimensions.

To test the effectiveness of additive HSGPs compared to RFF GPs, we experimented with the "large" datasets used in Delbridge et al. (2020) and found that DOEBEs consisting of additive HSGPs may often outperform DOEBEs consisting of RFF GPs — more details are available in Appendix E.

## 5 Experiments

We provide three different experiments in the main text, with additional experiments in the appendices. In the first experiment (Section 5.1), we compare ensembles of several different basis expansions, showing that the best-performing model varies widely. In the second experiment (Section 5.2), we show how model collapse between static and dynamic models can occur, and how the model introduced in Section 4.2 alleviates it. Finally, we show that E-DOEBE can effectively ensemble methods that are static and dynamic, and of different basis expansions (Section 5.3).

The metrics we use are the normalized mean square error (nMSE), and the predictive log-likelihood (PLL). The nMSE is defined to be the MSE of $y_t$ with the predictive mean, divided by the variance of $y_{1:T}$. In particular, at time $t$, the nMSE is

$$\text{nMSE}_t = \frac{\sum_{\tau=1}^{t} (\mu_{y_\tau} - y_\tau)^2}{t \cdot \text{Var}(y_{1:T})}$$

The predictive log-likelihood (PLL), which is the average value of $\log p(y_{t+1}|\mathbf{X}_{1:t}, \mathbf{y}_{1:t})$, i.e., $\text{PLL}_t = \sum_{\tau=1}^{t} \log p(y_{\tau+1}|\mathbf{X}_{1:\tau}, \mathbf{y}_{1:\tau})/t$.

Across all experiments, we use several publicly available datasets, ranging in both size and the number of features. A table of dataset statistics is available in Table 1. Friedman #1 and Friedman #2 are synthetic datasets designed to be highly nonlinear, and notably, are i.i.d. The Elevators dataset is related to controlling the elevators on an aircraft. The SARCOS dataset uses simulations of a robotic arm, and Kuka #1 is a similar real dataset from physical experiments. CaData is comprised of California housing data, and the task of CPU Small is to predict a type of CPU usage from system properties.

All hyperparameter optimization was done on the first 1,000 samples of each dataset; since we already assume access, each dataset was additionally standardized in both $\mathbf{x}$ and $y$ using the statistics of the first 1,000 samples. We follow Lu et al. (2022) in setting a weight to 0 when it falls below the threshold of $10^{-16}$.

---

[6]In this context, by "high-dimensional," we mean $D \gtrsim 20$. This is similar to its usage in, e.g., Binois & Wycoff (2022).

| Dataset Name | Number of Samples | Dimensionality $d$ |
|---|---|---|
| Friedman #1 (Friedman, 1991) | 40,000 | 10 |
| Friedman #2 (Friedman, 1991) | 40,000 | 4 |
| Elevators (Torgo) | 16,599 | 17 |
| SARCOS (Rasmussen & Williams, 2005) | 44,484 | 21 |
| Kuka #1 (Meier et al., 2014) | 197,920 | 21 |
| CaData (Pace & Barry, 1997) | 20,640 | 8 |
| CPU Small (Delve Developers, 1996) | 8,192 | 12 |

Table 1: Dataset statistics, including the number of samples, the number of features, and the original source. In addition to the original sources above, several of these datasets were curated by the UCI Machine Learning Repository (Kelly et al.) or LibSVM (Chang & Lin, 2011).

## 5.1 Comparing Different Basis Expansions

To show that having a wide variety of basis expansion models available is useful, we test several model types on each dataset of Table 1. Additionally, we test static and dynamic versions of models to compare performance.

Models used for comparison include an additive HSGP model [**(D)OE-HSGP**], an RFF GP [**(D)OE-RFF**], an ensemble of quadratic, cubic, and quartic polynomials with additive structure [**(D)OE-Poly**], linear regression [**(D)OE-Linear**], and a one-layer RBF network [**(D)OE-RBF**]. Apart from additional hyperparameter tuning in an ARD kernel, the (D)OE-RFF model is identical to the (D)IE-GP of Lu et al. (2022).

For RFF GPs, 50 Fourier features were used (so $F = 2 \times 50$), and for HSGPs, $\lfloor 100/D \rfloor$ features were used for each dimension (so $F \lesssim 100$). In either case, an SE-ARD kernel was used. For RBF networks, 100 locations were initialized via K-means and then optimized with empirical Bayes, as well as ARD length scales. For all models except RBF networks, ensembles were created by the process proposed in Section 4.1 — for RBF networks, the computation of the Hessian was too expensive, and so parameters were randomly perturbed by white Gaussian noise with variance $10^{-3}$ instead.

For dynamic models, $\sigma_{\mathrm{rw}}^2$ was set to $10^{-3}$ following Lu et al. (2022). The initial values of $\sigma_{\boldsymbol{\theta}}^2$ and $\sigma_{\epsilon}^2$ were 1.0 and 0.25, respectively. Optimization was performed with Adam (Kingma & Ba, 2014).

**Results** Results of the average nMSE and PLL can be found in Fig. 2. We provide full plots of (online) nMSE and PLL as a function of samples seen in Appendix F. We find that the best-performing class of models differs dramatically from dataset to dataset. In particular, in terms of both nMSE and PLL, HSGPs, RFF GPs, and RBF networks all attain the best performance on at least one dataset. This reinforces the idea that ensembling with respect to several different models is beneficial, as there is not a single method consistently outperforming the others.

Additionally, as expected, dynamic models can dramatically outperform static models in specific cases (for example, on SARCOS and Kuka #1), but attain a worse PLL on datasets where the data is reasonably i.i.d. (for example, Friedman #1).

As expected, when an additive structure is a reasonable approximation, additive HSGP methods outperform RFF GPs, for example on Kuka #1 and CaData. The RFF GP approximation rarely performs particularly poorly, making it a consistently "good" estimator, and achieves the highest PLL on Friedman #2, SARCOS, and CPU Small. However, it is also at times outperformed by simpler methods, such as the RBF network, highlighting the potential benefits of using diverse basis expansions.

**Key Takeaways** Key takeaways from this experiment include: (1) neither dynamic nor static methods are strictly superior across all settings, (2) no single basis expansion is superior across all datasets, and (3) RFF GPs consistently provide good performance, but this performance can often be improved upon by using other basis expansions.

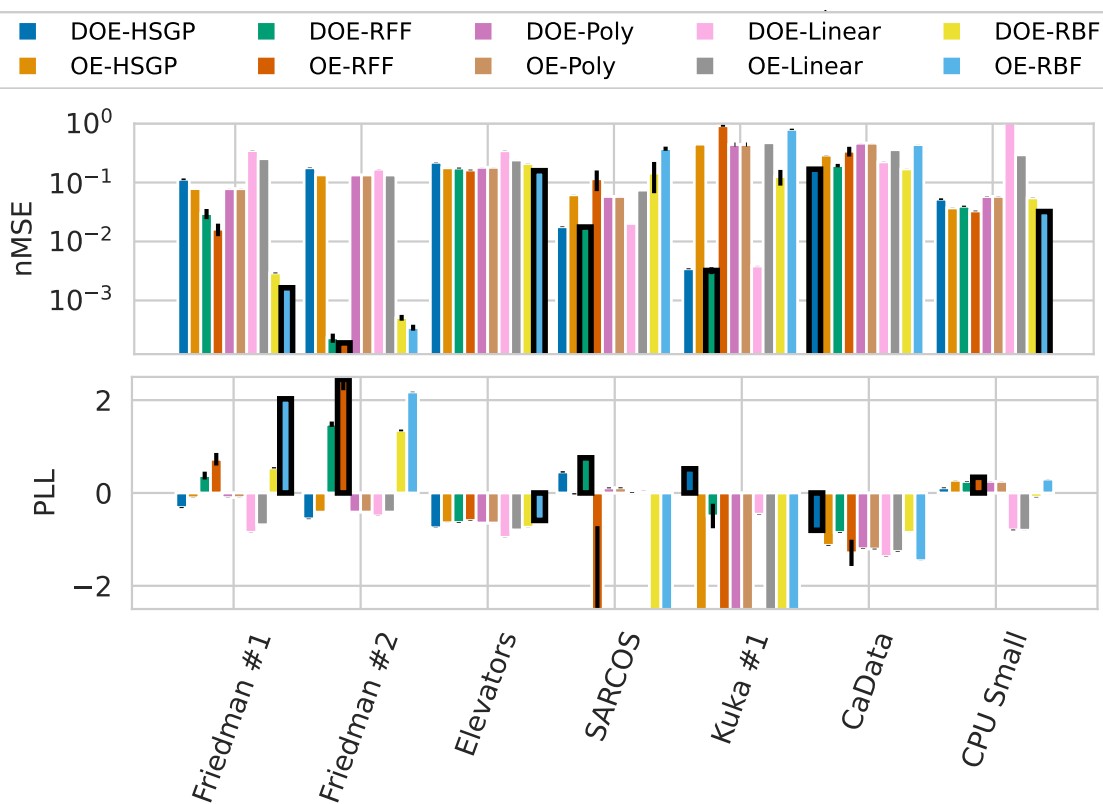

Figure 2: Results of Experiment 1. Pictured are the nMSE (lower is better) and PLL (higher is better) with error bars denoting one standard deviation over 10 random trials. The best-performing method on each dataset and metric is highlighted with bold edges — to preserve readability, the nMSE axis was bounded at $+1.0$ and the PLL axis is bound at $\pm 2.5$, even if points extend past this.

## 5.2 The Necessity of Ensembles of Dynamic Ensembles

In this experiment, we show that the E-DOEBE model proposed in Section 4.2 can indeed prevent premature collapse of BMA weights. While this premature collapse of BMA weights does not seem typical in real datasets, it is not hard to illustrate its possibility, even on real datasets with high-performing methods.

As a constructive example, we can create an ensemble of additive HSGPs on the Kuka #1 dataset, where dynamic models performed significantly better in Section 5.1. In particular, we created an ensemble of two additive HSGPs where the first model is dynamic (in particular, $\sigma_{\mathrm{rw}}^{(1)^2} = 10^{-3}$) and the second model is static (i.e., $\sigma_{\mathrm{rw}}^{(2)^2} = 0$). The ensemble hyperparameters were fit with empirical Bayes, with the initial length scale values being the vector of ones. Then, the resulting ensemble was trained online as a DOEBE and as an E-DOEBE, with $\delta = 10^{-2}$. Note that in this carefully controlled setting, each basis expansion is entirely deterministic given the hyperparameters, so the results are purely deterministic and cannot be attributed to poor random seeds.

The resulting weights, pictured in Fig. 3, demonstrate that premature collapse of BMA weights can be a problem. Numerically, the log-likelihood of the E-DOEBE model is dramatically better than that of the DOEBE model (Table 2), showing this collapse can be catastrophic.

This issue can be partially averted by eliminating the threshold of $10^{-16}$ when ensembling. Indeed, in this example, the weights reach a minimum of approximately $10^{-72}$. However, with any finite precision arithmetic, there is always the potential for this type of collapse to occur due to numerical underflow. It is

trivial to construct such examples by generating the first $N_1$ samples with $\sigma_{\text{rw}}^{(m)} = 0$ until weight collapse occurs, and the rest of the dataset with $\sigma_{\text{rw}}^{(m)} > 0$.

**Key Takeaway**   The key takeaway of this experiment is that an ensemble of dynamic and static models can catastrophically collapse — even when the discrepancy in performance along the entire dataset is large — and that the E-DOEBE approach proposed in Section 4.2 can avoid this collapse.

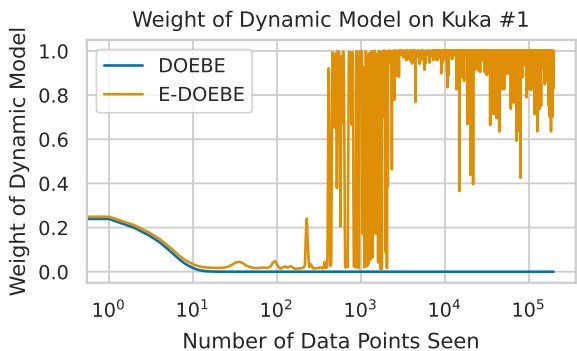

Figure 3: Results of Experiment 2. Pictured is the 10-sample moving average of the BMA weight corresponding the dynamic additive HSGP when trained as a DOEBE and E-DOEBE.

| Method | Predictive Log-Likelihood |
|--------|---------------------------|
| DOEBE | -403.41 |
| E-DOEBE | **0.55** |

Table 2: Predictive log-likelihood of DOEBE and E-DOEBE models in Experiment 2 (higher is better).

## 5.3   E-DOEBE Outperforms Other Methods

The ultimate goal of the E-DOEBE model is to combine static and dynamic models of several different types. To do so, we repeat the experiments of Section 5.1 while comparing to an E-DOEBE model. We restrict our attention to static and dynamic versions of the three best-performing families of models in Experiment 1 ((D)OE-HSGP, (D)OE-RFF, and (D)OE-RBF), and an E-DOEBE ensemble containing all of them. The E-DOEBE model is created with $\delta = 10^{-2}$, which was not tuned. Results are presented in Fig. 4. Full (online) nMSE and PLL plots from each experiment can be found in Appendix G.

As desired, the E-DOEBE model can effectively ensemble dynamic and static models of different basis expansions. Across all experiments, the E-DOEBE model performs the best in terms of PLL, and is the best in terms of NMSE for all but one dataset (Friedman #2).

**Key Takeaway**   The E-DOEBE can effectively ensemble several different ensembles of high-performing basis expansions, resulting in consistently better performance than any single method.

## 5.4   Additional Experiments

We undertook several additional experiments which are omitted from the main text but included in appendices. The first such experiment is a simple streaming classification task using the Laplace approximation outlined in Section 3.4 (Appendix A). We also perform an experiment validating that the sampling technique of Section 4.1 can be beneficial (Appendix C), and a small ablation study on the $\delta$ parameter of the E-DOEBE (Appendix D). We also compared the (D)OE-RFF and (D)OE-HSGP methods on additional datasets appearing in the GP literature (Appendix E). Finally, we provide some comments and experiments regarding a variant of (D)OE-RFF where the frequencies are optimized (Appendix H).

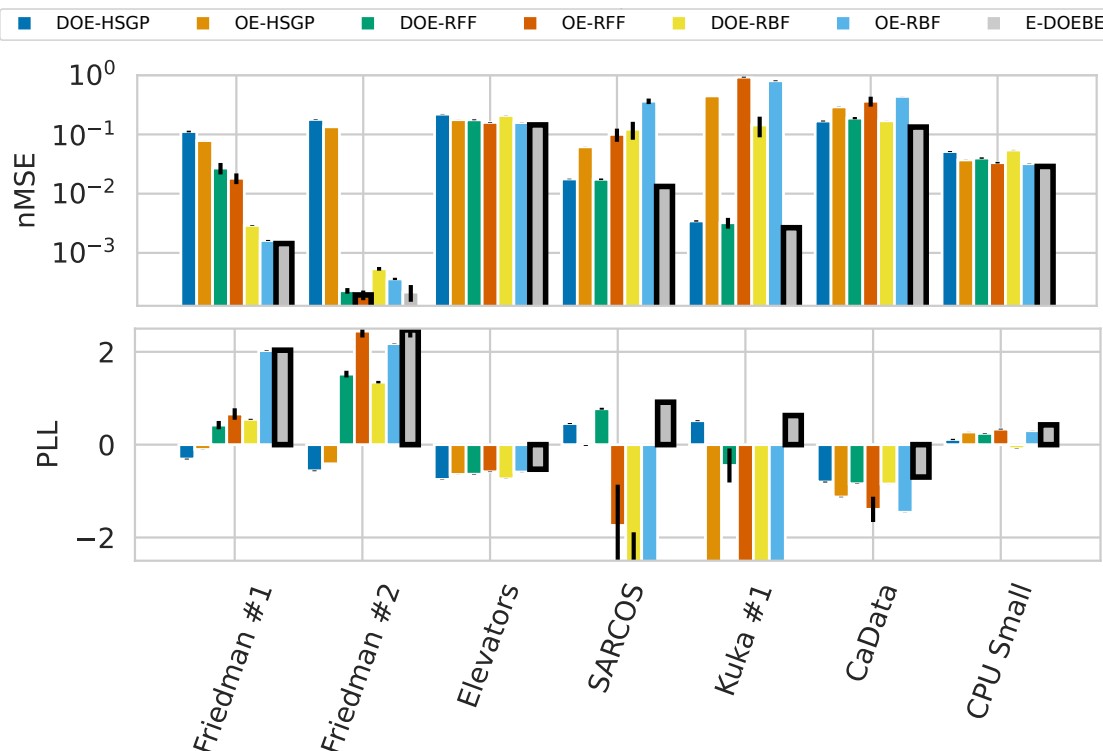

Figure 4: Results of Experiment 3. Pictured are the nMSE (lower is better) and PLL (higher is better) with error bars denoting one standard deviation over 10 random trials. The best-performing method on each dataset and metric is highlighted with bold edges — to preserve readability, the nMSE axis was bounded at +1.0 and the PLL axis is bound at ±2.5, even if points extend past this.

## 6 Conclusion & Future Directions

In this paper, we showed that recent progress in online prediction using RFF GPs can be expanded to arbitrary linear basis expansions. This included several basis expansions that outperform RFF GPs on real and synthetic datasets. We show how, in a simple framework, different linear basis expansions can be ensembled together, increasing ensemble diversity. While several popular choices of basis expansions were used, it would be interesting to expand the tests even further, particularly with splines.

We also showed that premature collapsing of BMA weights can be an issue in online ensembling. We introduced the E-DOEBE model, which alleviates this issue, and showed its effectiveness. However, this meta-ensembling may be seen as adding a complex bandage to BMA rather than addressing the underlying issues. Further research may explore how to incorporate other Bayesian ensembling methods, such as Bayesian (hierarchical) stacking (Yao et al., 2018; 2022).

While we provide guidance on how to initialize ensembles given a set of basis expansions, deciding which basis expansions to use is an important open topic. A naïve approach would be to expand on the existing use of the marginal likelihood for model selection, but this may be "unsafe" when using different basis expansions (Llorente et al., 2023) and therefore requires care. We additionally provided several ideas for inference with non-Gaussian likelihoods, for example, for classification tasks. Determining which, if any, of these tasks is superior to the Laplace approximation is another interesting topic for future study.

Finally, it could be beneficial to modify or add new basis expansions in the online setting. Indeed, recent progress in GPs has worked towards selecting and adapting kernels online to great benefit (Duvenaud et al., 2013; Shin et al., 2023). If such techniques could be adapted to DOEBE, it could eliminate the pre-training period, and allow for adapting the domain of approximations when new data arrives.

**Acknowledgments**

This work was supported by the National Science Foundation under Award 2212506.

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

# A   Classification Experiment

In the main body of the text, we focused on the empirical performance of (E-)DOEBE on Gaussian regression problems. While there are several possible improvements to classification (or other non-Gaussian) problems, as outlined in Section 3.4, we perform a simple experiment on the common toy dataset "banana" (Rätsch et al., 2001).[7]

The banana dataset is notably not exchangeable, with data being loaded from left to right. Based on model knowledge and the experimental results in Section 5.1, we would expect dynamic models to perform better on the non-exchangeable dataset, and static models to perform better on a shuffled (i.e., i.i.d.) version. We first experiment with the shuffled version of the dataset, and then with the unshuffled version. Models in each experiment are identical to those of Section 5.3, except using logistic likelihood via the Laplace approximation.

## A.1   Shuffled Data

By shuffling the data, we expect it to be i.i.d., in which case we expect static models to perform well and the classification error to be steadily decreasing. We show the classification error as a function of the number of data seen and an example of the predictive distribution for the OE-RFF, both pictured in Fig. 5

The data is obviously non-additive, so the HSGPs do not perform particularly well. The dataset is quite simple and, consistent with our regression experiments on simple i.i.d. datasets, the RBF networks perform quite well. Notably, the E-DOEBE consistently performs as well or better than the best-performing model. Finally, we note that the results with RFF models are comparable to those reported in Lu et al. (2022) for IE-GPs.

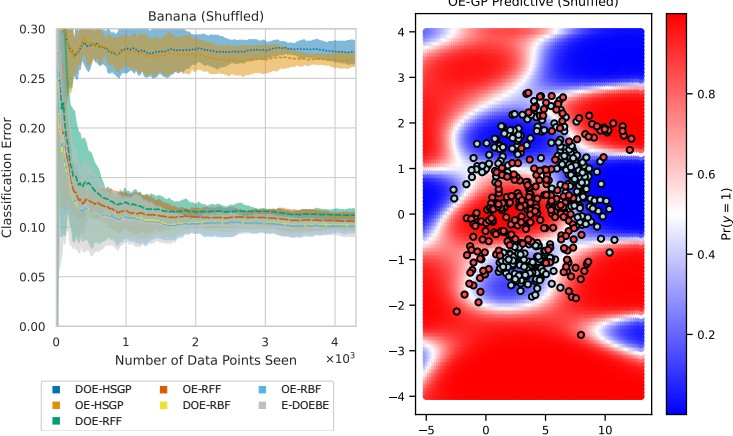

Figure 5: **(Left)** Classification errors for the shuffled banana dataset. Lines indicate the mean over 5 trials, with shaded regions denoting $\pm 1$ standard deviation. **(Right)** The predictive distribution of an OE-RFF ensemble after training on the shuffled banana dataset.

## A.2   Unshuffled Data

We also experimented with the case where the data is not shuffled. In this case, the data is presented from left-to-right, as visualized in Fig. 6. The resulting classification errors are pictured in Fig. 7 and are comparable to several other works performing streaming GP classification (Stanton et al., 2021; Bui et al., 2017), where classification errors are at or around 10%.

Again consistent with our results for regression tasks, the dynamic models perform better in this non-exchangeable setting. Intuitively, the RBF models perform quite poorly as the locations $\boldsymbol{\mu}_k$ are only trained

---

[7]The original link to the dataset is no longer available, but a copy is available online at Diethe (2015).

on the first 1,000 points, which are not representative of the entire dataset. The RFF models still perform well, and interestingly, the DOE-HSGP model performs competitively as well. This may be because "local" additivity is a more realistic condition.

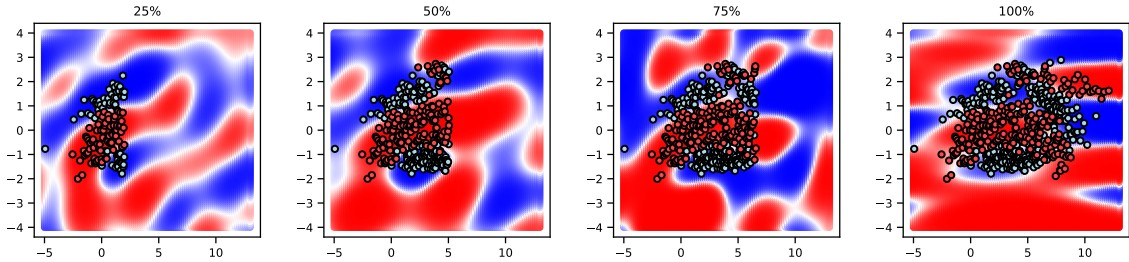

Figure 6: The predictive distribution of an OE-RFF ensemble after observing the first 25%, 50%, 75%, and 100% of the data, respectively.

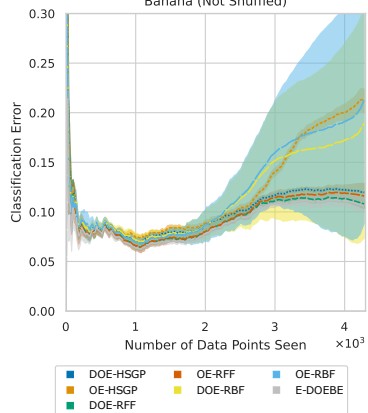

Figure 7: Classification errors for the unshuffled banana dataset. Lines indicate the mean over 10 trials, with shaded regions denoting ±1 standard deviation.

# B   Proof of Theorem 1

In this appendix, we provide the proof for the online regret analysis bound given in Theorem 1. The proof here is essentially the same as the proof in Lu et al. (2022, Section 9) (and in turn, the proof of Theorem 2.2 in Kakade & Ng (2004)). It is provided for notational consistency and completeness.

*Proof of Theorem 1.* The proof consists of three steps: (1) bounding the regret of the OEBE to any given expert, (2) bounding the regret of any expert with respect to a variational distribution, and (3) bounding the regret of the variational distribution with respect to a fixed strategy.

**Step 1**   For step (1), note that by definition of the BMA weights, we have that

$$\frac{w_{\tau-1}^{(m)}}{w_\tau^{(m)}} = \frac{\exp(-\ell_\tau)}{\exp(-l_\tau^{(m)})},$$

Since the initial weights are $w_0^{(m)} = 1/M$, this implies

$$\exp\left(-\sum_{\tau=1}^T \ell_\tau + \sum_{\tau=1}^T l_\tau^{(m)}\right) = \frac{w_0^{(m)}}{w_1^{(m)}} \frac{w_1^{(m)}}{w_2^{(m)}} \cdots \frac{w_{T-1}^{(m)}}{w_T^{(m)}} = \frac{1}{Mw_T^{(m)}}.$$

Together, these bound the regret of the OEBE with respect to expert $m$,

$$\sum_{\tau=1}^T \ell_\tau - \sum_{\tau=1}^T l_\tau^{(m)} = \log M + \log w_t^{(m)} \leq \log M, \tag{33}$$

since $\log w_t^{(m)} \leq 0$ by construction.

**Step 2**   For step (2), we introduce a variational distribution $q(\boldsymbol{\theta}^{(m)}) = \mathcal{N}(\boldsymbol{\theta}^{(m)}; \boldsymbol{\theta}_*^{(m)}, \xi^2 \boldsymbol{I}_{F_m})$ with variational parameter $\xi^2$. We then bound the regret of expert $m$ with respect to the expected loss under $q(\cdot)$,

$$\sum_{\tau=1}^T l_\tau^{(m)} - \mathbb{E}_q[\mathcal{L}_{\boldsymbol{\theta}^{(m)}}],$$

where $\mathcal{L}_{\boldsymbol{\theta}^{(m)}} = p(y_{1:T}|\boldsymbol{\theta}^{(m)}, \mathbf{x}_{1:T})$. By Bayes's rule, we get that $\sum_{\tau=1}^T l_\tau^{(m)} = -\log p(y_{1:T}|\mathbf{x}_{1:T})$ (i.e., the marginal probability of $y_{1:t}$ given $\mathbf{x}_{1:t}$ under model $m$), and therefore

$$\sum_{\tau=1}^T l_\tau^{(m)} - \mathbb{E}_q[\mathcal{L}_{\boldsymbol{\theta}^{(m)}}] = \int q(\boldsymbol{\theta}^{(m)}) \log \frac{p(y_{1:T}|\boldsymbol{\theta}^{(m)}, \mathbf{x}_{1:T})}{p(y_{1:T}|\mathbf{x}_{1:T})} \, d\boldsymbol{\theta}^{(m)}$$

since $\log p(y_{1:T}|\mathbf{x}_{1:T})$ does not depend on $\boldsymbol{\theta}$. By the definition of conditional densities, we rewrite

$$\frac{p(y_{1:T}|\boldsymbol{\theta}^{(m)}, \mathbf{x}_{1:T})}{p(y_{1:T}|\mathbf{x}_{1:T})} = \frac{p(\boldsymbol{\theta}^{(m)}|y_{1:T}, \mathbf{x}_{1:T})}{p(\boldsymbol{\theta}^{(m)})}.$$

We will then multiply by $1 = q(\boldsymbol{\theta}^{(m)})/q(\boldsymbol{\theta}^{(m)})$ and rearrange, to get

$$\sum_{\tau=1}^T l_\tau^{(m)} - \mathbb{E}_q[\mathcal{L}_{\boldsymbol{\theta}^{(m)}}] = \int q(\boldsymbol{\theta}^{(m)}) \frac{q(\boldsymbol{\theta}^{(m)})}{p(\boldsymbol{\theta}^{(m)})} \, d\boldsymbol{\theta}^{(m)} - \int q(\boldsymbol{\theta}^{(m)}) \frac{q(\boldsymbol{\theta}^{(m)})}{p(\boldsymbol{\theta}^{(m)}|y_{1:T}, \mathbf{x}_{1:T})} \, d\boldsymbol{\theta}^{(m)}.$$

Recognizing each term as a Kullback Liebler (KL) divergence, this is clearly bounded by $D_{\mathrm{KL}}\left(q(\boldsymbol{\theta}^{(m)}) \,\|\, p(\boldsymbol{\theta}^{(m)})\right)$ by non-negativity of the KL divergence, resulting in the bound

$$\sum_{\tau=1}^T l_\tau^{(m)} - \mathbb{E}_q[\mathcal{L}_{\boldsymbol{\theta}^{(m)}}] \leq D_{\mathrm{KL}}\left(q(\boldsymbol{\theta}^{(m)}) \,\|\, p(\boldsymbol{\theta}^{(m)})\right)$$

$$= F_m \log \sigma_{\boldsymbol{\theta}}^{(m)} + \frac{\|\boldsymbol{\theta}_*^{(m)}\|^2 + F_m \xi^2}{2\sigma_{\boldsymbol{\theta}}^{(m)^2}} - \frac{F_m}{2}(1 - 2\log\xi). \tag{34}$$

**Step 3** Finally, for (3), we will bound the difference

$$\mathbb{E}_q[\mathcal{L}_{\boldsymbol{\theta}^{(m)}}] - \mathcal{L}_{\boldsymbol{\theta}_*^{(m)}}.$$

By taking a Taylor series expansion of $\mathcal{L}(\cdot, y_\tau)$ centered at $z_{t,*} = \phi^{(m)}(\mathbf{x}_\tau)^\top \boldsymbol{\theta}_*^{(m)}$, we find

$$\mathbb{E}_q[\mathcal{L}_{\boldsymbol{\theta}^{(m)}}] - \mathcal{L}_{\boldsymbol{\theta}_*^{(m)}} = \mathbb{E}_q\left[\frac{d^2}{dz_\tau^2}\mathcal{L}(g(z_\tau); y_\tau)\frac{(z_\tau - z_{\tau,*})^2}{2}\right]$$

for $z_\tau = \phi^{(m)}(\mathbf{x}_\tau)^\top \boldsymbol{\theta}^{(m)}$ and some function $g(\cdot)$. By invoking our assumption about boundedness of the second derivative of $\mathcal{L}(\cdot; y_\tau)$, we see

$$\mathbb{E}_q\left[\frac{d^2}{dz_\tau^2}\mathcal{L}(g(z_\tau); y_\tau)\frac{(z_\tau - z_{\tau,*})^2}{2}\right] \leq \mathbb{E}_q\left[c\frac{(z_\tau - z_{\tau,*})^2}{2}\right],$$

and by invoking our assumption that $\|\phi^{(m)}(\mathbf{x})\|^2 \leq 1$, we see

$$\mathbb{E}_q[\mathcal{L}_{\boldsymbol{\theta}^{(m)}}] - \mathcal{L}_{\boldsymbol{\theta}_*^{(m)}} = \mathbb{E}_q\left[c\frac{(z_\tau - z_{\tau,*})^2}{2}\right] \leq \frac{c\xi^2}{2}. \tag{35}$$

With each step complete, we must now combine them to finish our proof. By combining Eqs. (33) to (35), we find

$$\sum_{\tau=1}^{T}\ell_t - \sum_{\tau=1}^{T}\mathcal{L}(\phi^{(m)}(\mathbf{x}_\tau)^\top\boldsymbol{\theta}_*^{(m)}; y_\tau) \leq \frac{Tc\xi^2}{2} + F_m\log\sigma_{\boldsymbol{\theta}} + \frac{\|\boldsymbol{\theta}_*^{(m)}\|^2 + n\xi^2}{2\sigma_{\boldsymbol{\theta}}^{(m)2}} + \frac{F_m}{2}(1 - 2\log\xi). \tag{36}$$

After minimizing with respect to the variational parameter $\xi^2$ and simplifying, the theorem is proven. $\square$

## C  Experiments with Sampling Hyperparameters for Ensembles

To show the benefits of using a Laplace approximation to sample hyperparameters, we repeat the experiments of Section 5.1 on the Elevators, SARCOS, and CaData datasets using just DOE-HSGP and DOE-RFF models, creating two copies: one using the type-II MLE hyperparameters, and the other using the Laplace approximation of the marginal likelihood. The experiment consisted of 100 trials where 10 samples were drawn from each Laplace approximation. For RFF GPs, the frequencies of the Monte Carlo approximation were fixed to be the same for each trial. The method with better performance was determined via the one-sided Wilcoxon signed-rank test in the relevant direction. Results can be found in Table 3.

| Method | Predictive Log-Likelihood | | | Normalized Mean Square Error | | |
|---|---|---|---|---|---|---|
| | Elevators | SARCOS | CaData | Elevators | SARCOS | CaData |
| DOE-HSGP-MLE | $-0.753 \pm 0.000$ | $0.421 \pm 0.000$ | $0.081 \pm 0.000$ | $0.221 \pm 0.000$ | $\mathbf{0.017 \pm 0.000}$ | $0.055 \pm 0.000$ |
| DOE-HSGP-Sample | $\mathbf{-0.748 \pm 0.003}$ | $\mathbf{0.466 \pm 0.010}$ | $\mathbf{0.120 \pm 0.010}$ | $\mathbf{0.219 \pm 0.001}$ | $0.018 \pm 0.000$ | $\mathbf{0.052 \pm 0.001}$ |
| DOE-RFF-MLE | $-0.640 \pm 0.007$ | $0.756 \pm 0.018$ | $0.243 \pm 0.009$ | $0.178 \pm 0.003$ | $0.018 \pm 0.001$ | $0.040 \pm 0.001$ |
| DOE-RFF-Sample | $\mathbf{-0.639 \pm 0.007}$ | $\mathbf{0.766 \pm 0.019}$ | $\mathbf{0.247 \pm 0.009}$ | $\mathbf{0.177 \pm 0.004}$ | $0.018 \pm 0.001$ | $\mathbf{0.040 \pm 0.002}$ |

Table 3: Predictive likelihood (higher is better) and normalized MSE (lower is better) of type-II MLE and Laplace-approximated initialization, plus/minus one standard deviation over 100 trials. Bolded entries denote superior performance significant at the $p = 0.05$ level according to a one-sided Wilcoxon rank-sum test.

Overall, the benefit of sampling from the posterior on these datasets appears to be small but significant. In particular, in terms of predictive log-likelihood (which DOEBE targets), the Laplace approximation initialization never performs worse.

# D Sensitivity of E-DOEBE With Respect to $\delta$

In this section, we test the performance of the E-DOEBE model used in Section 5.3 across different values of $\delta \in [0.01, 0.05, 0.1, 0.2]$. We ran the E-DOEBE model on each dataset once, and recorded the PLL. Results can be found in Fig. 8. We conclude that a value of $\delta \simeq 0.05$ typically performs well, but performance is reasonably similar for all $0.01 \le \delta \le 0.2$.

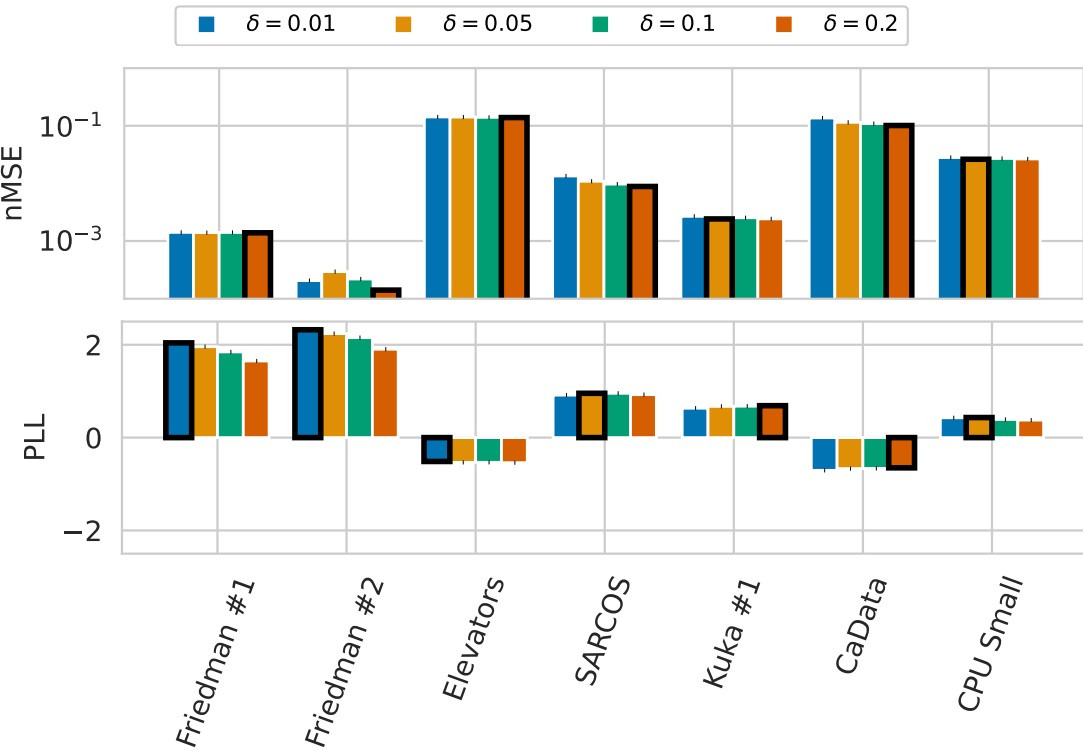

Figure 8: Results of the $\delta$ ablation experiment. Pictured are the nMSE (lower is better) and PLL (higher is better) over one trial. The best-performing method on each dataset and metric is highlighted with bold edges.

# E   Further Experiments on Large Datasets

To support our assertion that additive HSGPs may be superior to RFF GPs in medium-high dimension, we perform experiments on the UCI datasets (Kelly et al.) used in Delbridge et al. (2020). A summary of each dataset used can be found in Table 4.

| Dataset Name | Number of Samples | Dimensionality $d$ |
|:---:|:---:|:---:|
| autos | 159 | 25 |
| servo | 167 | 4 |
| machine | 209 | 7 |
| yacht | 308 | 6 |
| autompg | 392 | 7 |
| housing | 506 | 13 |
| stock | 536 | 11 |
| energy | 768 | 8 |
| concrete | 1,030 | 8 |
| airfoil | 1,503 | 5 |
| gas | 2,565 | 128 |
| skillcraft | 3,338 | 19 |
| sml | 4,137 | 26 |
| pol | 15,000 | 26 |
| bike | 17,379 | 17 |
| kin40k | 40,000 | 8 |

Table 4: Dataset statistics, including the number of samples and the number of features for datasets used in Delbridge et al. (2020). All datasets are available on the UCI Machine Learning Repository (Kelly et al.).

In particular, we use the (D)OE-HSGP and (D)OE-RFF models of Section 5.1, with the following modifications: the number of Fourier features was set to 250 (so $F = 2 \times 250$), and $\lfloor 500/D \rfloor$ features were used for each dimension (so $F \lesssim 500$). Because the datasets are often much shorter, empirical Bayes estimates are taken over the first 100 points only. Dimensions of the input vectors which were identically zero for all samples were removed. Results can be found in Fig. 9.

Overall, we find that (D)OE-HSGPs outperform (D)OE-RFFs on several tasks, with varying input dimension $D$ and length $N$. This includes several datasets which would typically be considered high-dimensional in the GP literature, for example, the gas dataset ($D = 128$) and the sml dataset ($D = 26$).

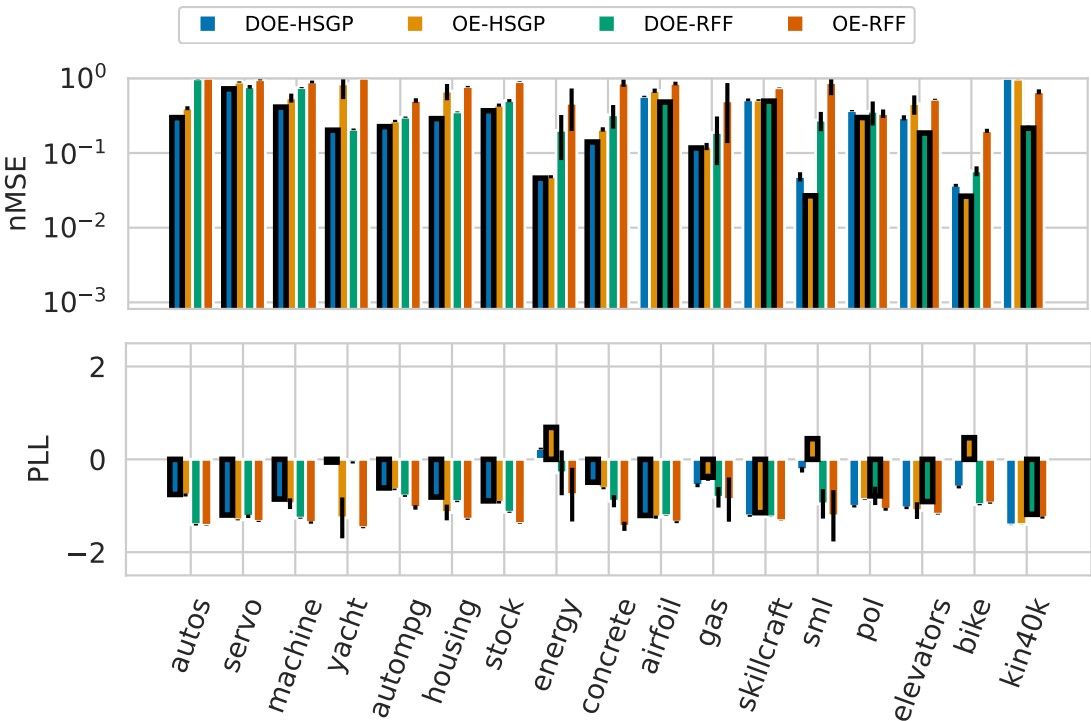

Figure 9: Results of additional experiments on UCI datasets. Pictured are the nMSE (lower is better) and PLL (higher is better) with error bars denoting one standard deviation over 10 random trials. The best-performing method on each dataset and metric is highlighted with bold edges — to preserve readability, the nMSE axis was bounded at +1.0 and the PLL axis is bound at ±2.5, even if points extend past this.

## F  Results by Dataset for Experiment 1

Here, we provide the full nMSE and PLL plots from the experiment in Section 5.1.

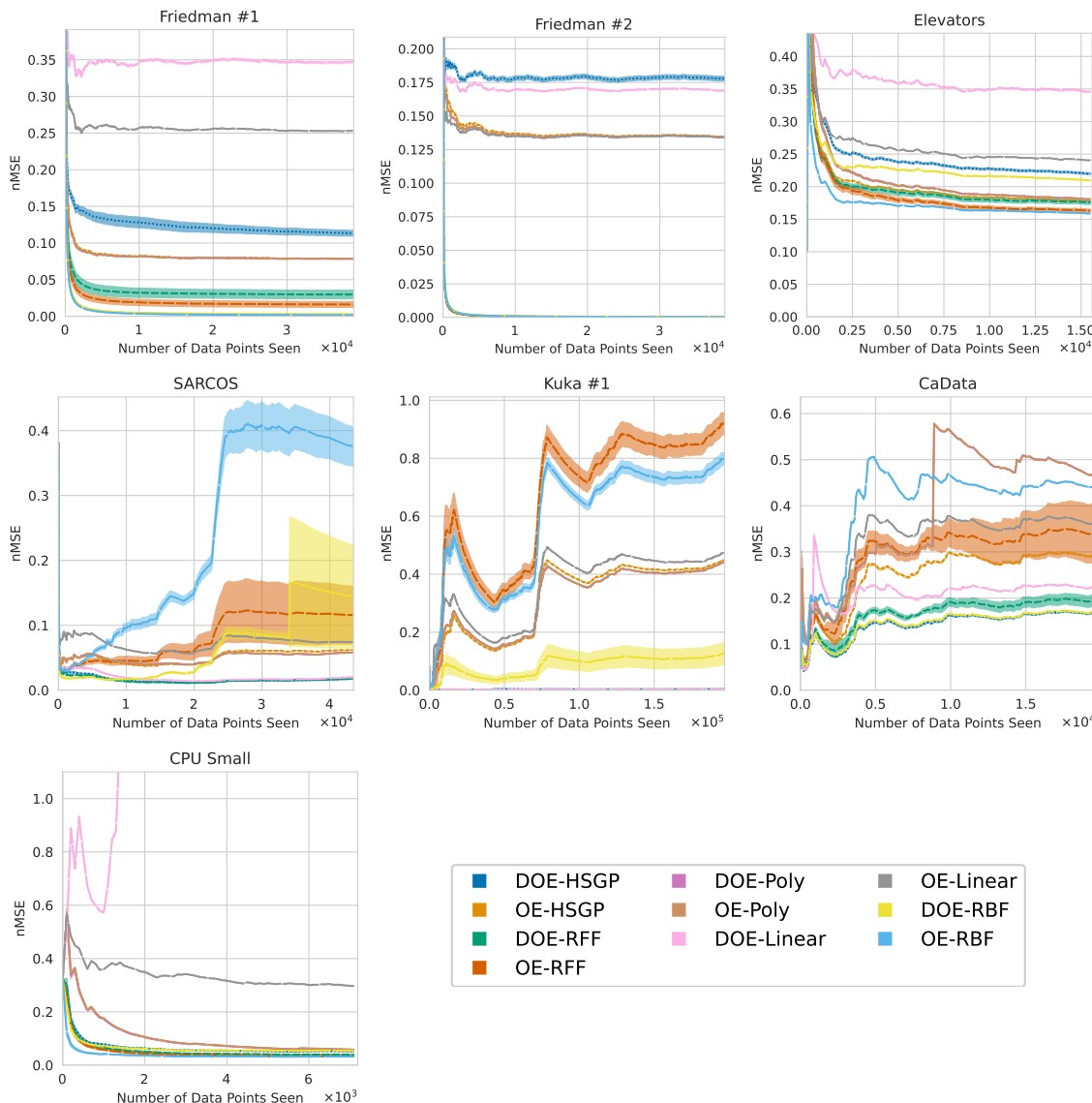

Figure 10: nMSE plots of each method tested in the experiment of Section 5.1. Lines indicate the mean over 10 trials, with shaded regions denoting ±1 standard deviation.

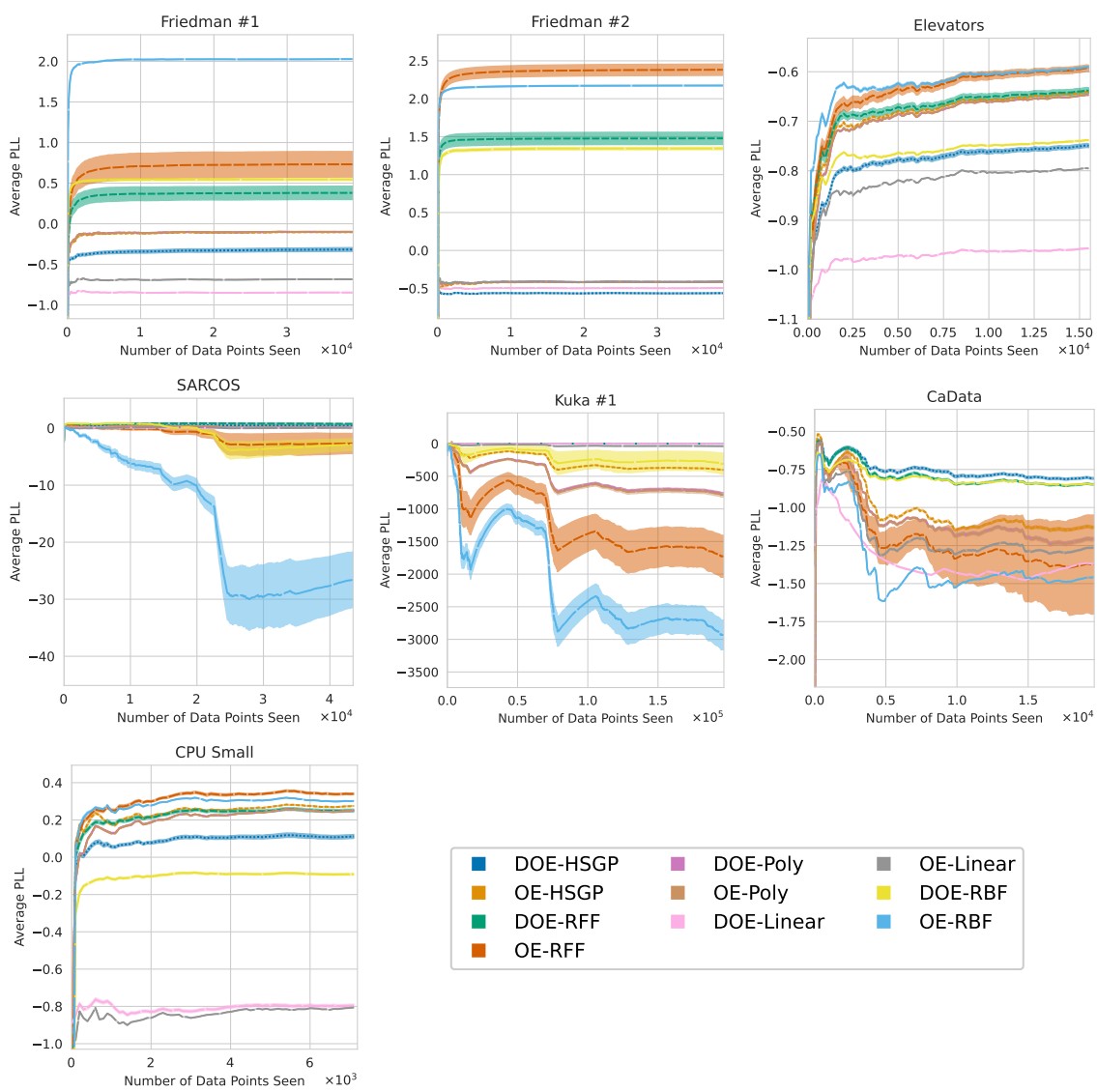

Figure 11: PLL plots of each method tested in the experiment of Section 5.1. Lines indicate the mean over 10 trials, with shaded regions denoting ±1 standard deviation.

## G   Results by Dataset for Experiment 3

Here, we provide the full nMSE plots from the experiment in Section 5.3.

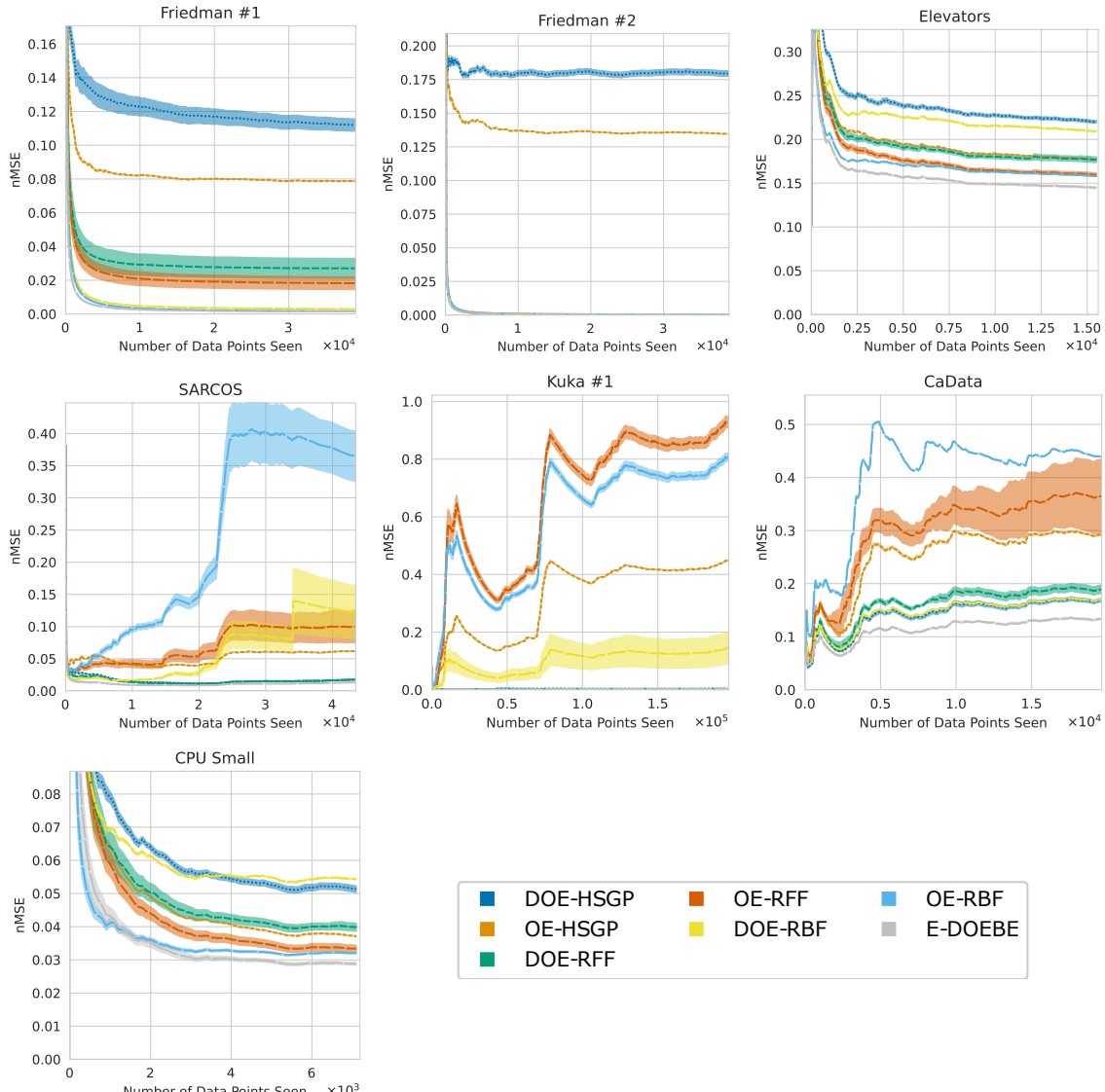

Figure 12: nMSE plots of each method tested in the experiment of Section 5.3. Lines indicate the mean over 10 trials, with shaded regions denoting $\pm 1$ standard deviation.

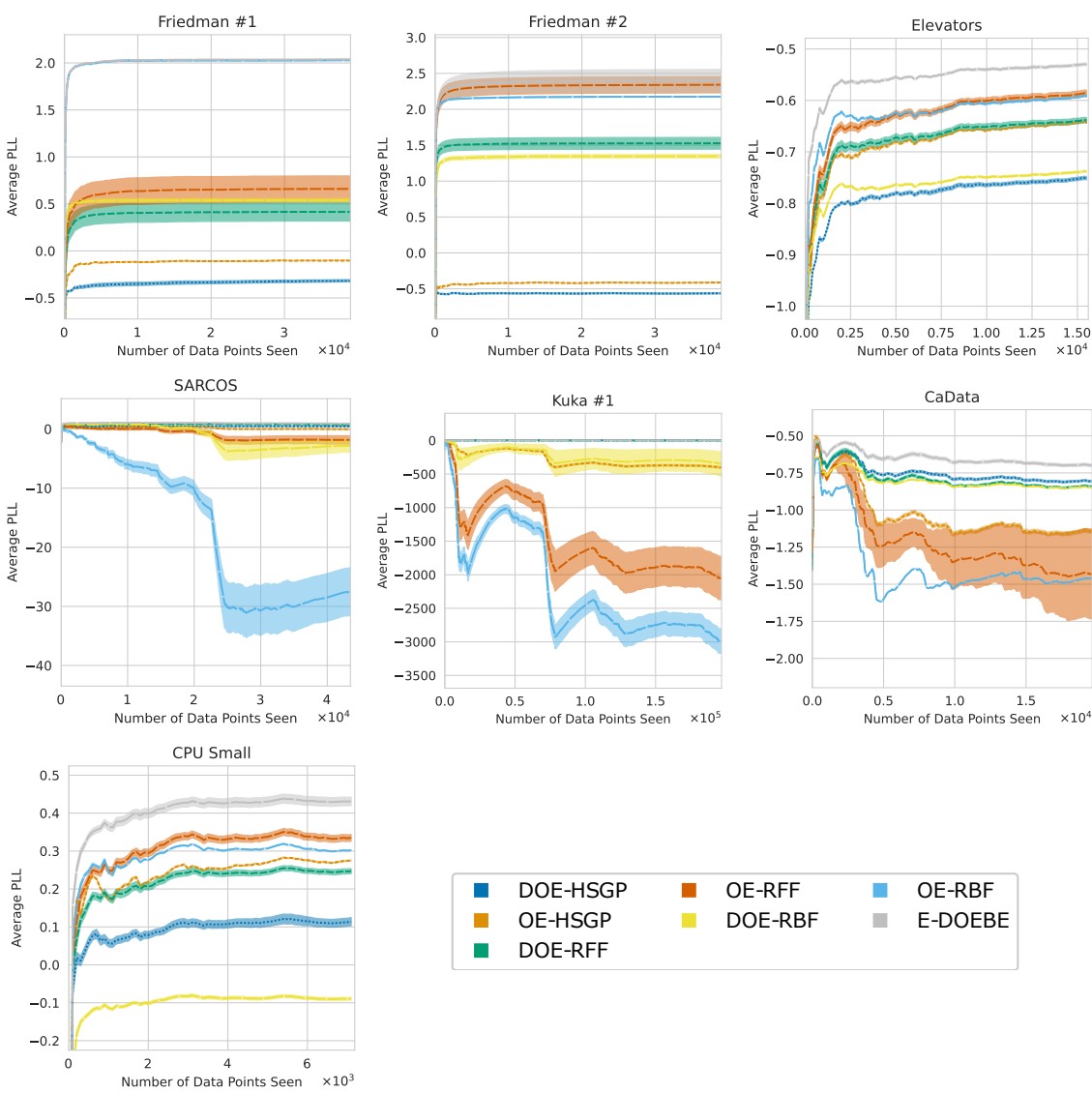

Figure 13: PLL plots of each method tested in the experiment of Section 5.3. Lines indicate the mean over 10 trials, with shaded regions denoting ±1 standard deviation.

# H   Optimizing Fourier Features

Our framework and implementation make it simple to additionally optimize the Fourier features $\boldsymbol{s}_m$ of the RFF model, which we will refer to as the *optimized Fourier features (OFF)* model. Such an optimization is not novel and was indeed explored in the original paper introducing RFF for GPs (Lázaro-Gredilla et al., 2010). The idea of optimizing spectral points also forms the basis of the popular spectral mixture kernel (Wilson & Adams, 2013), which uses a Gaussian mixture in the spectral domain instead of a particle representation.

In some sense, optimizing frequencies defeats the main motivation of RFF GPs; we can no longer claim that the results converge to any particular GP as $m \to \infty$, since we are no longer approximating Eq. (19) for a pre-specified kernel. Indeed, one can view the resulting model as a one-hidden-layer neural network, with activation function $\phi_{\text{RFF}}(\cdot)$; likewise, we expect the results to inherit both benefits of neural networks (e.g. flexibility) and drawbacks (e.g. harsh overfitting).

We additionally note that the resulting model bears a significant resemblance to the Gaussian RBF basis expansion model. As will become evident in our results, there is likewise much similarity in their performance: on "simple" datasets, optimizing frequencies provides large benefits. On more complex — or dynamic — datasets, it is instead prone to overfitting.

For experiments, we repeat Section 5.1 using the DOE-RFF model, and a version with optimized frequencies (DOE-OFF). Hyperparameters for both were sampled with the "Gaussian" strategy, as the number of hyperparameters is too large in the optimized version for the Laplace approximation. Results can be found in Fig. 14.

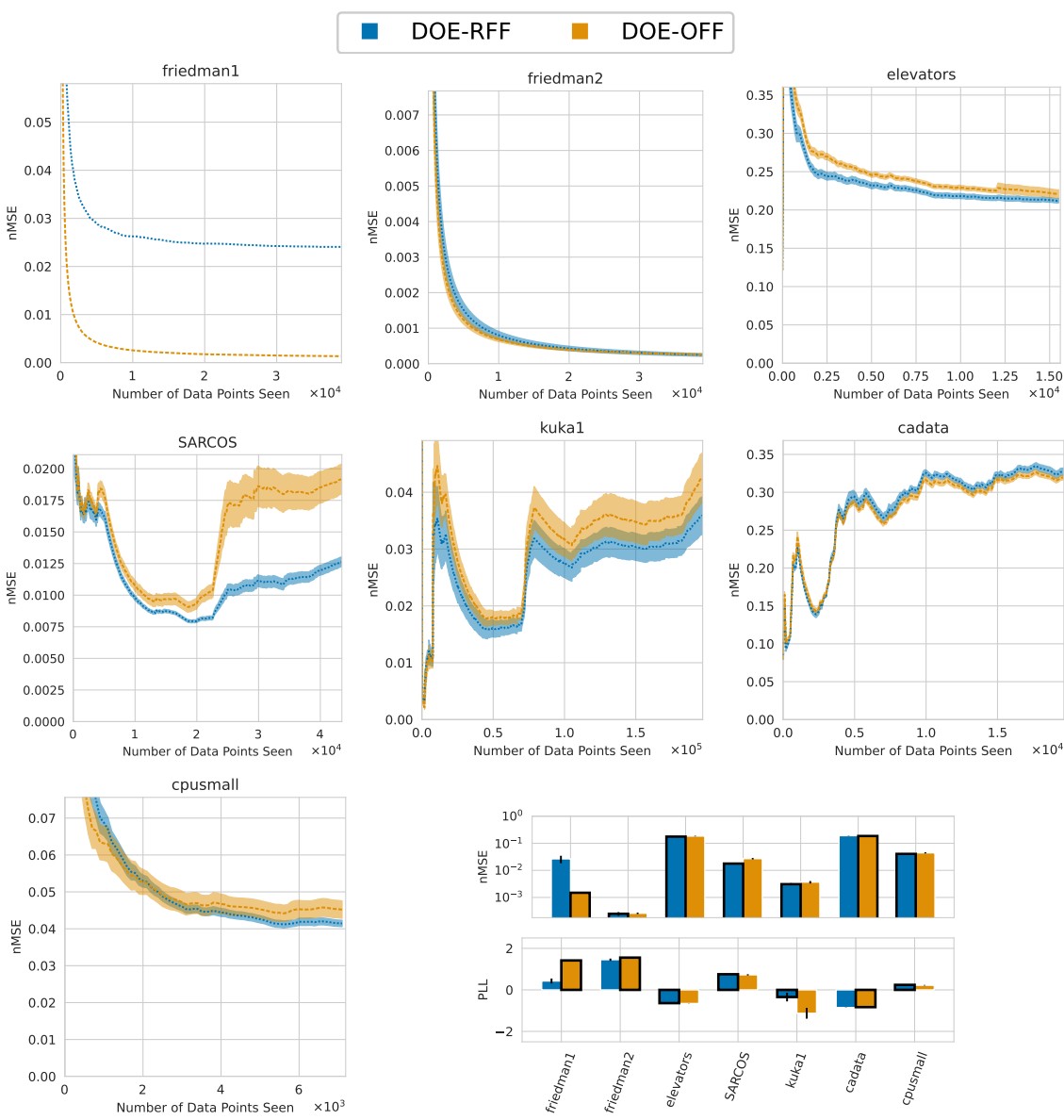

Figure 14: **(Continuous Plots)** nMSE plots of random and optimized Fourier features. Lines indicate the mean over 10 trials, with shaded regions denoting ±1 standard deviation. **(Bar Plots)** Summaries of the nMSE (lower is better) and PLL (higher is better) with error bars denoting one standard deviation over 10 random trials. The best-performing method on each dataset and metric is highlighted with bold edges — to preserve readability, the nMSE axis was bounded at +1.0 and the PLL axis is bound at ±2.5, even if points extend past this.

