# OpenReview forum: "Dynamic Online Ensembles of Basis Expansions"
_TMLR — Accepted by TMLR_

### Review · Reviewer_jUqT · 2024-02-24

**Summary Of Contributions:**

Random (Fourier) features are a widely used method for approximating Gaussian process inference, and have recently been used to ensemble GPs in an online manner. The authors show how such methods can be used and extended in the setting where any basis expansion model is used (e.g. Hilbert space GP). Empirically, they test various features and basis expansions and find that different settings and datasets result in different rankings of performance. Bayesian model averaging is discussed and investigated. Ensembling of dynamic models and static models is also investigated.

**Audience:**

Yes

**Claims And Evidence:**

Yes

**Requested Changes:**

- There is an error in equation (15)-(17). See Equation (2.19) of Rasmussen and Williams book. The authors provided the posterior predictive distribution of the noise free prediction f(X*), but they wrote that this was the posterior predictive distribution of the noiseless distribution y*. I believe what they really want to write is the noise free predictive distribution f(X*). This is in contrast with the earlier finite dimensional setting, where predict and correct updates require the predictive distribution with noise for y*.
- The authors incorrectly state that inference requires inversion of an $N \times N$ matrix. What is really required is to \emph{solve} a linear system, which also scales cubically in $N$, but requires less memory than inversion, has a smaller constant, and is more numerically stable. Again in section 2.3.
- In section 2.3.1, you could mention any of the classical variance estimates of RFF. E.g. "On the Error of Random Fourier Features", Sutherland and Schneider. I am not sure if you can also provide variance estimates for the more recent Hilbert Space GP. If these are available, it would be great to compare and contrast, since the claim empirically seems to be that sometimes RFF is better than HSGP but not consistently, at least in your current context.
- The last part of the Figure 3 caption does not make any sense and contains some weird whitespace.

Minor:
- First page, BMA (Bayesian Machine Averaging) is used without introducing the acronym.
- Second paragraph of 2.1.2. Citation for  “empirical Bayes,” “type-II maximum likelihood estimation,” and “evidence maximization” seems oddly recent. Maybe also Rasmussen and Williams?
- Third paragraph of 2.1.2. Is this an instance of conjugate prior updates? Perhaps that could be worth mentioning.
- No date on the Torgo dataset in table 1. Is this known?

**Strengths And Weaknesses:**

**strengths:**

Several approaches to GPs are describeded (infinite dimensional, RFF, HSGP, ...), as are approaches to hyperparameter fitting (point estimate, sampling from Laplace approx of marignal likelihood) and Bayesian model averaging. The text is of appropriate length and detail.

All of these techniques are evaluated empirically in a suite of appropriate empirical benchmarks.

The authors do a good job in acknowledging limitations of their study and potential future work, e.g. "We introduced the E-DOEBE model, which alleviates this issue, and showed its effectiveness. However, this meta-ensembling may be seen as adding a complex bandage to BMA rather than addressing the underlying issues".

**weaknesses:**

The results are entirely empirical. While this is okay, it would be nice if the authors could at least mention some of the existing theoretical results so that they can relate their empirical findings to the theory (see requested changes below).

There are some mostly minor errors (see requested changes below).

---

> ### Author Response · Authors · 2024-03-25
> **Response to Reviewer jUqT**
>
> Thank you for your helpful comments on improving our manuscript. We've made several changes to the manuscript based on your comments, which we outline below.
>
> **Requested Changes**
> > - There is an error in equation (15)-(17). See Equation (2.19) of Rasmussen and Williams book. The authors provided the posterior predictive distribution of the noise free prediction f(X*), but they wrote that this was the posterior predictive distribution of the noiseless distribution y*. I believe what they really want to write is the noise free predictive distribution f(X*). This is in contrast with the earlier finite dimensional setting, where predict and correct updates require the predictive distribution with noise for y*.
>
> This is correct, thank you. We have corrected the exposition and notation to reflect that this is the predictive distribution of $f(X_*)$ and that the predictive distribution of $y_*$ can be obtained by adding the noise covariance.
>
>
> > - The authors incorrectly state that inference requires inversion of an $N \times N$ matrix. What is really required is to *solve* a linear system, which also scales cubically in $N$, but requires less memory than inversion, has a smaller constant, and is more numerically stable. Again in section 2.3.
>
> Thank you for pointing this out. Our emphasis was on the cubic scaling in time, and it is relatively common to colloquially state that inference requires inversion, but it is important to point out that actual implementations are more numerically stable and more memory efficient than inverting the training covariance matrix. We have changed the exposition to reflect this.
>
> > - In section 2.3.1, you could mention any of the classical variance estimates of RFF. E.g. "On the Error of Random Fourier Features", Sutherland and Schneider. I am not sure if you can also provide variance estimates for the more recent Hilbert Space GP. If these are available, it would be great to compare and contrast, since the claim empirically seems to be that sometimes RFF is better than HSGP but not consistently, at least in your current context.
>
> We have included some discussion in a new section about online regret bounds (Section 3.5). This includes a discussion on how high probability error bounds of RFF GPs are used to prove regret bounds with respect to the underlying RKHS and how the results of Sutherland and Schneider can improve said bounds.
>
> Analogous results for uniform convergence of HSGPs are available (though not variance estimates, as it is a deterministic method), which we believe can be used to prove similar regret bounds. However, we suspect much of the error in HSGPs is instead related to the use of a GAM approximation, rather than HSGPs vs. RFF GPs.
>
> > - The last part of the Figure 3 caption does not make any sense and contains some weird whitespace.
>
> We agree that the formatting of Figure 3/Table 2 was poor and that it caused confusion with other reviewers as well. We have updated the caption and changed the formatting to be inline.
>
> > - First page, BMA (Bayesian Machine Averaging) is used without introducing the acronym.
>
> Thank you, fixed.
>
> > - Second paragraph of 2.1.2. Citation for “empirical Bayes,” “type-II maximum likelihood estimation,” and “evidence maximization” seems oddly recent. Maybe also Rasmussen and Williams?
>
> The reasoning for the use of (Theodoridis, 2020) is that all three names (empirical Bayes, type-II MLE, and evidence maximization) are explicitly mentioned. We have added the additional sentence "Empirical Bayes has a long history of being effectively used for Bayesian linear basis expansions and Gaussian process regression", with references to Bishop and Rasmussen and Williams.
>
> > - Third paragraph of 2.1.2. Is this an instance of conjugate prior updates? Perhaps that could be worth mentioning.
>
> Yes, this is the result of integrating out the variance term when the conjugate normal inverse-gamma model is used. We have mentioned this and added a reference to a recent extension of IE-GPs which performs this conjugate update.
>
> > - No date on the Torgo dataset in table 1. Is this known?
>
> There is no concrete date listed on the website, and after a brief literature search, we were unable to find a consistent date listed. We have chosen to add a note to the bibliography reference stating "Last Accessed March 19, 2024" instead.

---

> > ### Comment · Reviewer_jUqT · 2024-04-04
> >
> > Thanks for responding to my comments. I think you have addressed all of my concerns.

---

### Review · Reviewer_j5fF · 2024-03-18

**Summary Of Contributions:**

This paper uses a hierarchical ensemble of basis expanded linear models for online learning. The weights in the ensemble are given by the previous steps’ predictive log likelihood, while collapse of the ensemble to focus on a single model is prevented by using a structured matrix. Furthermore, noise is added to each model’s weights to ensure that it can update effectively to new data. Overall, these additions make each model a type of linear state space model, while the Bayesian model averaging piece is also a state space model in some sense.

The main novelty of the work seems to be that it expands Lu et al, ‘22’s efforts from random fourier features to other Bayesian linear models.

**Audience:**

Yes

**Claims And Evidence:**

No

**Requested Changes:**

Comments / corrections:

-	Figure 3 / Table 2: It may be better presentation wise to inline both (for the table, use the tabular environment). As currently placed together, it looks like the captions are running on to each other.

-	Figure 3: show +- 2sd of mean for these trials.

-	Figure 2: please don’t trim the NLL for sarcos / kuka #1. I personally tend to think that each dataset should be its own small plot for this reason; however, I understand that this is more time consuming. This would enable different scales for each dataset.

-	Figures – remove “Results of Experiment []” from captioning.

-	Footnote 5: missing period at end.

-	Figure 6 – I believe what is marked as “S-DOEBE” should be “E-DOEBE”.

-	Presentation wise, I have a bit of trouble comparing the different online learning experiments, which all seem to be mixed up with a comparison of different basis expansions (quite unimportant).

      o	I would suggest that the authors pick one basis expansion they like and instead compare it against an explicitly defined IEGP with a similar enough kernel from Liu et al, ’22 as well as other online learning techniques using either GPs or linear models (e.g. Stanton et al, ’21, Yadav e al, ’21, Zhang et al, ’23).

-	It’s not especially clear to me why these ensembles should be better than a single Gaussian process / RFF, especially as the authors use the MMSE estimate with a single Gaussian distribution. So, by linearity, shouldn’t this mean that there’s really only one set of weights \theta_t?

      o	I think I understand that there’s differences in the additive setting when there are various different inputs for each model (each dimension). However, if the inputs are the same, shouldn’t there really only be one global set of weights?

Experiments (to me, these are of high importance):

-	I’d suggest that the authors focus primarily on the additive Hilbert space features and conduct experiments on high dimensional datasets, e.g. the ones from Delbridge et al, ’20.

-	I’d suggest that the authors try some sort of online classification problems like in some of the other online GP literature. A useful first approximation may be to turn classification into a regression problem via reformulating the likelihood like was done in Stanton et al, ’21.

        o	 Comparison could then be done to the IE-GP on the datasets from Liu et al, ’21.

-	Basic baselines that would be nice to include for the learning curves are the performance of non-online models trained on various amounts of data.

-	To alleviate my concern that the ensembles are not necessarily better than a single model, it would be good to study M = 1, 2, 3, …, 5 (or so) on the kuka dataset.

Code:

I took a quick look at the code and it is nicely documented with good comments. A few small comments here:

-	L72 of src/DOEBE/sdoebe.py: should ` final_log_w, log_ws` be the same vector?

-	L118 of src/DOEBE/doebe.py: why is the svd of the pseudo-inverse necessary? Shouldn’t it just be something like

```python
hess_matrix = jnp.block(hess)
u, s, _ = jnp.linalg.svd(hess_matrix)
inv_hess_factor = u @ jnp.diag(jnp.sqrt(1.0 / s))
```

Modulo potentially needing to clip s at zero…


References:

Liu et al, 2022, Incremental Ensemble Gaussian Processes.

Stanton et al, 2021, Kernel Interpolation for Scalable Gaussian Processes.

Yadav et al, 2021, Faster Kernel Interpolation for Gaussian Processes.

Zhang et al, 2023, Sequential Gaussian Processes for Online Learning of Nonstationary Functions.

**Strengths And Weaknesses:**

Strengths:

-	The ideas and their ideological histories are clearly presented and well explained. I really appreciate the longer length so that each concept can be clearly introduced.

-	The writing and summary are nicely done and the statement of contributions at the beginning summarizes the paper nicely.

-	I also like that the code was provided and in a pretty readable state.

Weaknesses:

-	Overall, there’s not a huge amount of novelty especially compared to Lu et al, ’22. The new method seems to be enabling just other basis expansions (many of which are severely dimensionality limited). The experimental results don’t currently suggest that these other options are a very good idea as RFFs require less tuning and as the authors state “it rarely performs particularly poorly”.

       o	Furthermore, this comes at a cost of not being able to handle non-Gaussian likelihoods very well which the IE-GP can handle.

-	To distinguish between static and dynamic models, there’s no real experiments (e.g. Bayesian optimization and active learning) which limits the “practical” applicability in some sense (more on this in the suggestions).


-	The hyperparameters seem to be fixed after the first 1000 data points. Including a periodic retraining or continuous optimization steps ought to help make even static models even more “dynamic”.

(review edited for viewing clarity)

---

> ### Author Response · Authors · 2024-03-25
> **Response to Reviewer J5fF [1/3]**
>
> Thank you for your detailed review and suggestions for improving our manuscript. We've made several changes to the manuscript based on your comments, which we outline below.
>
> **Weaknesses**
> >  - Overall, there’s not a huge amount of novelty especially compared to Lu et al, ’22. The new method seems to be enabling just other basis expansions (many of which are severely dimensionality limited). The experimental results don’t currently suggest that these other options are a very good idea as RFFs require less tuning and as the authors state “it rarely performs particularly poorly”.
>
> We agree that generalizing the results of Lu et al. to arbitrary basis expansions is the main contribution of our work. However, we highlight additional novelty in the E-DOEBE model, which shows how to ensemble several different types of basis expansions together. Further, the section on initialization is much further developed than in the work on IE-GPs. Additionally, one of the principal conclusions of the experimental section is that other basis expansions, for example, additive HSGPs, will at times outperform RFF GPs. We believe this to be a novel observation within this framework that is of practical importance to the community --- even if RFF GPs rarely perform particularly poorly, it is still often better to try other models, or include other models in an ensemble *a la* DOEBE.
>
> > Furthermore, this comes at a cost of not being able to handle non-Gaussian likelihoods very well which the IE-GP can handle.
>
> We respectfully disagree with this statement. The recursive estimation of linear models is exactly the theoretical underpinning of IE-GPs, and to handle classification, Lu et al. utilize the Laplace approximation. There is no reason the Laplace approximation cannot be applied to the general basis expansion case -- indeed, a standard way to introduce the Laplace approximation is for Bayesian logistic regression using linear models (see, e.g., Chapter 4 of Bishop). We recognize this connection may not be obvious to the reader, and have added a new section to the manuscript addressing non-Gaussian likelihoods (Section 3.4). This includes the Laplace approximation, as well as references to several other possible approaches, including posterior linearization and particle filtering.
>
> > - The hyperparameters seem to be fixed after the first 1000 data points. Including a periodic retraining or continuous optimization steps ought to help make even static models even more “dynamic”.
>
> We agree that this is a worthwhile improvement in some settings. Continuous optimization is made difficult as this corresponds to simultaneously learning a Bayesian model and changing the model being learned. To the best of our knowledge, such approaches where the measurement model of a Kalman filter is changed online are not available. Periodic retraining can be useful in some applications, but makes online learning much more difficult, in the sense that learning must be "paused" to pursue the expensive operation of retraining before continuing. We have added comments to the manuscript in Section 4.1 regarding this idea of periodic retraining, outlining when it might be useful.
>
> **Requested Changes**
> > - Figure 3 / Table 2: It may be better presentation wise to inline both (for the table, use the tabular environment). As currently placed together, it looks like the captions are running on to each other.
>
> Thank you for this suggestion. Indeed, the formatting of Figure 3/Table 2 seemed confusing to other reviewers as well. We have implemented your suggestion of presenting them inline.
>
> > - Figure 3: show +- 2sd of mean for these trials.
>
> Because this experiment is comprised of two basis expansions which are deterministic conditioned on their hyperparameters (HSGPs) trained via empirical Bayes, the results are deterministic and there are no error bars to show in the results. We have made a comment regarding this in the manuscript to avoid confusion.
>
> > - Figure 2: please don’t trim the NLL for sarcos / kuka #1. I personally tend to think that each dataset should be its own small plot for this reason; however, I understand that this is more time consuming. This would enable different scales for each dataset.
>
> We made the choice to present as is done currently primarily to consolidate information and not inundate the reader with too many plots, but we appreciate the concern for untrimmed NLLs. In the initial version of the manuscript, plots of the online nMSE for each experiment/dataset appeared in Appendices B and C (Appendices E and F, in the revised version); we have since added NLL plots as well and a reference within the manuscript to point the reader to un-trimmed plots.
>
> > Figures - remove "Results of Experiment \[\]" from captioning.
>
> We prefer to keep this in the captioning, as we believe it quickly disambiguates which figure corresponds to what section for a reader skimming through the manuscript.

---

> > ### Comment · Reviewer_j5fF · 2024-04-06
> > **Thanks for the comments**
> >
> > Thank you for the updated paper and comments. Several of the revised figures look much better. I still have a few comments and concerns:
> >
> > Laplace approximations: Thanks for the section on here, I tend to think that it would be useful to actually just do the classification experiment inside of this framework, rather than merely writing it out. It doesn't have to be a particularly challenging problem, but the streaming bananas classification from Stanton et al and Bui et al is probably a good place to start.
> >
> > Periodic retraining / online learning: Indeed, these can be somewhat tricky to get right, but are often critical for many practical applications so I still tend to think that not running these experiments is a weakness of this paper even if one can theoretically motivate it somewhat in your framework. Related works tend to do at least some gradient updates - Stanton et al continue training throughout, while Bui et al and Maddox et al manage to get some retraining in the online learning process (but Bui considers primarily large batch online learning).
> >
> >
> > References:
> >
> > Bui et al, 2017, Streaming sparse Gaussian process approximations.
> >
> > Stanton et al, 2021, Kernel Interpolation for Scalable Gaussian Processes.
> >
> > Maddox et al, 2021, Conditioning Sparse Variational Gaussian Processes for Online Decision-making

---

> > > ### Author Response · Authors · 2024-04-11
> > > **Reply to Reviewer j5fF [1/2]**
> > >
> > > Thank you once again for your continued engagement in the discussion.
> > >
> > > > Laplace approximations: Thanks for the section on here, I tend to think that it would be useful to actually just do the classification experiment inside of this framework, rather than merely writing it out. It doesn't have to be a particularly challenging problem, but the streaming bananas classification from Stanton et al and Bui et al is probably a good place to start.
> > >
> > > Thanks for the suggestion. We maintain that if classification is of particular interest, it is likely worthwhile to investigate approximations other than Laplace. Nonetheless, we have included an experiment on the banana dataset (now in Appendix A). As the banana dataset is obviously non-additive, RFF GPs outperform additive HSGPs in the ensemble, and the results are essentially comparable to IE-GPs as a result -- which are in turn fairly comparable to the results in Stanton et al. and Bui et al.
> > >
> > > > Periodic retraining / online learning: Indeed, these can be somewhat tricky to get right, but are often critical for many practical applications so I still tend to think that not running these experiments is a weakness of this paper even if one can theoretically motivate it somewhat in your framework. Related works tend to do at least some gradient updates - Stanton et al continue training throughout, while Bui et al and Maddox et al manage to get some retraining in the online learning process (but Bui considers primarily large batch online learning).
> > >
> > > We acknowledge other works include online gradient-based learning, but we would like to note that they often are in a variational framework (for example, both the papers of Bui et al. and Maddox et al. which were mentioned), where optimization seems more natural and appropriate. In contrast, online gradient ascent of the marginal likelihood while simultaneously doing Bayesian recursions is not obviously well-posed. As stressed in the text, how exactly to implement this (e.g., with periodic retraining) seems application-dependent, and so we prefer to omit it.
> > >
> > > > MMSE estimates: In the case when the input basis expansions are not the same, isn't that just equivalent to a "stacked" model with the basis features `\Phi = [\phi^{(1)}(x_t), \cdots, \phi^{(M)}(x_t)]` and then possibly some weighting? The prediction doesn't seem nonlinear at all to me as w can be rolled into the likelihood (e.g. like a fixed noise heteroscedastic loss)...
> > >
> > > Indeed, the prediction is equivalent to the predictive distribution of this "stacked" model, but this does not reflect how each model is trained. Training each model separately and then ensembling in the predictive step is not the same as rolling $w$ into the likelihood; for example, consider one basis expansion $x \mapsto [ 1 \, x ]$ and another $x \mapsto [ x^2 ]$. Clearly, if the models related to these basis expansions are trained separately (as is done in DOEBE), one does not generally obtain the same result as when using the basis expansion $x \mapsto [1 \, x \, x^2]$. Regardless of this, the pertinent aspect is that predictions are highly non-linear in $x$. One could in principle use the entire Gaussian mixture instead of the MMSE estimate, but as stressed in the manuscript, the answers do not differ much after a few thousand data points. We therefore follow Lu et al. in using the MMSE estimate since it's a good approximation in this case and the implementation is simpler.
> > >
> > > > And, from a skim of the proof of the regret analysis in Kakade and Ng, their analysis exploits the single linear model case (the extension in Liu et al also basically relies on this too..)
> > >
> > > We believe there may be confusion in the theorems and proofs of Kakade and Ng; what they refer to as "Bayesian model averaging" is different than in the current context -- it is not related to ensembling and just refers to prediction using the posterior predictive distribution. The part where BMA enters the current regret analysis is in what we refer to as "Step 1" in our proof (which, again, is essentially the same as the proof of Lu et al.). Kakade and Ng bound a regret on the log-likelihood of the posterior predictive of a basis expansion with respect to any fixed value of $\theta_*$. The additional step of Lu et al. is to bound the log-likelihood of the ensemble to that of the posterior predictive for any given basis expansion (Equation (33) in our manuscript).

---

> > > ### Author Response · Authors · 2024-04-11
> > > **Reply to Reviewer j5fF [2/2]**
> > >
> > > > UCI datasets from Delbridge et al: I had more meant the ultra-high dimensional ones (olivetti and coepra) but thank you for these experiments. Unfortunately, I'm a bit confused here as there's no comparison to Delbridge et al on the same metrics (and I can't directly compare without digging into your code to get MSE which they report or to have an evaluation on NMSE from their method). I was mostly asking about this to try to get at if there's aperformance gap between fourier feature approaches and the full GPs.
> > >
> > > Thanks for the clarification. The performance gap between the random Fourier Feature GPs and the full GP is highly dependent on the number of Fourier features (as explored in-depth in \[1\] and \[2\]). Note that the analysis of \[1\] and \[2\] carries directly since the recursions in DOEBE are exact, i.e., they provide the same result as batch learning. A direct comparison to the results of Delbridge et al. is not simple, namely because they do not deploy the GPs in an online setting, which is the reason for differing metrics. From a theoretical perspective, high probability regret bounds are available relating the OE-RFF (i.e., IE-GP) and a full GP with the same kernel (e.g., Theorem 1 of Lu et al., and discussion in Section 3.5 about how they can be improved).
> > >
> > > > Indeed, thanks for the pointer. It still would be good to study how many pieces are in the ensemble (is two required for good performance or do we need all five?)
> > >
> > > If we are to assume i.i.d. data, in practice one expert is probably sufficient; the issue is picking the "correct" expert. This is not simple at initialization, particularly when using different basis expansions where evidence comparison may be considered "unsafe." In this sense, the BMA ensembling can be seen as a safer, principled alternative to pre-hoc model selection.
> > >
> > > We furthermore believe that the results of such an experiment would be intimately related to the dataset, models used, and their initializations. We therefore do not believe such results would be easily generalizable and would not feel comfortable making empirical claims about the number of experts needed.
> > >
> > > **References**
> > >
> > > \[1\] Lázaro-Gredilla, M., Quinonero-Candela, J., Rasmussen, C. E., & Figueiras-Vidal, A. R. (2010). Sparse spectrum Gaussian process regression. _The Journal of Machine Learning Research_, _11_, 1865-1881.
> > >
> > > \[2\] Sutherland, D. J., & Schneider, J. (2015). On the error of random Fourier features. arXiv preprint arXiv:1506.02785.

---

> ### Author Response · Authors · 2024-03-25
> **Response to Reviewer j5fF [2/3]**
>
> > - Footnote 5: missing period at end.
> > - Figure 6 – I believe what is marked as “S-DOEBE” should be “E-DOEBE”.
>
> Thank you for both of these corrections, which we have implemented.
>
> > - Presentation wise, I have a bit of trouble comparing the different online learning experiments, which all seem to be mixed up with a comparison of different basis expansions (quite unimportant).
>      I would suggest that the authors pick one basis expansion they like and instead compare it against an explicitly defined IEGP with a similar enough kernel from Liu et al, ’22 as well as other online learning techniques using either GPs or linear models (e.g. Stanton et al, ’21, Yadav e al, ’21, Zhang et al, ’23).
>
> The goal of Experiment 1 was to compare different basis expansions, illustrating the performance of each, with the takeaway that no single expansion could be determined as superior across all datasets. In Experiment 3, we take the suggested approach of choosing a subset of expansions that perform well in Experiment 1 (namely, RBF Networks, RFF GPs, and additive HSGPs), with the goal of showing E-DOEBE outperforms any single basis expansion.
>
> The kernel used for RFF GPs is fairly similar to those chosen in the IE-GP paper. In particular, the main difference is that we train the length scales of an ARD kernel rather than rely on a dictionary of fixed length scales.
>
> > - It’s not especially clear to me why these ensembles should be better than a single Gaussian process / RFF, especially as the authors use the MMSE estimate with a single Gaussian distribution. So, by linearity, shouldn’t this mean that there’s really only one set of weights \theta_t?
>     I think I understand that there’s differences in the additive setting when there are various different inputs for each model (each dimension). However, if the inputs are the same, shouldn’t there really only be one global set of weights?
>
> Even when using the MMSE estimate $y_t = \sum w^{(m)}\_t \mu^{(m)}\_{y\_t}$, there is nonlinearity imposed by the basis expansions, i.e., $y_t = \sum w^{(m)}_t \phi^{(m)}(x_t) \theta^{(m)}_t$ . Since the $\phi^{(m)}(\cdot)$ are different, the $\theta^{(m)}$ are all different. For example, in many of our experiments, $\theta^{(m)}$ is not even the same dimension for each $m$. Accordingly, if we are to view this prediction as arising from a "single set of weights," it is actually with respect to the vector $\theta_t = [\theta^{(1)} \cdots \theta^{(M)}].$
>
> On a related note, we have included a brief discussion of regret bounds, which bound the Bayesian loss of the online-learned ensemble to any member with a fixed set of parameters $\theta_*$.
>
> > - I’d suggest that the authors focus primarily on the additive Hilbert space features and conduct experiments on high dimensional datasets, e.g. the ones from Delbridge et al, ’20.
>
> We prefer to keep the current structure of exposition, where we first test several different basis expansions, show that ensembling static and dynamic models is not straightforward, and finally show the E-DOEBE performs well across our test suite. However, we have added a new appendix with experiments on the UCI datasets from Delbridge et al. (2020).
>
> We would like to note that several of the datasets used in the experimental section would already be considered "high dimensional" to many in the GP community, e.g., the SARCOS and Kuka #1 datasets ($d = 21$) (see e.g., Tables 1 and 2 of Ref. \[1\]) . We have added a footnote to avoid any confusion on the meaning of "high dimensional."
>
> > I’d suggest that the authors try some sort of online classification problems like in some of the other online GP literature. A useful first approximation may be to turn classification into a regression problem via reformulating the likelihood like was done in Stanton et al, ’21.
>
> While we have added a section on the possibility of non-Gaussian likelihoods, we prefer to focus on regression scenarios in our experiments. Accordingly, we do not make claims about performance in classification tasks in the manuscript. We believe that the issue of classification is an interesting aspect of future work, as there are potentially many ideas in the filtering literature to improve upon the Laplace approximation as used in IE-GPs.

---

> > ### Comment · Reviewer_j5fF · 2024-04-06
> > **Comments**
> >
> > MMSE estimates: In the case when the input basis expansions are not the same, isn't that just equivalent to a "stacked" model with the basis features `\Phi = [\phi^{(1)}(x_t), \cdots, \phi^{(M)}(x_t)]` and then possibly some weighting?  The prediction doesn't seem nonlinear at all to me as w can be rolled into the likelihood (e.g. like a fixed noise heteroscedastic loss)...
> >
> > - And, from a skim of the proof of the regret analysis in Kakade and Ng, their analysis exploits the single linear model case (the extension in Liu et al also basically relies on this too..)
> >
> > UCI datasets from Delbridge et al: I had more meant the ultra-high dimensional ones (olivetti and coepra) but thank you for these experiments. Unfortunately, I'm a bit confused here as there's no comparison to Delbridge et al on the same metrics (and I can't directly compare without digging into your code to get MSE which they report or to have an evaluation on NMSE from their method). I was mostly asking about this to try to get at if there's aperformance gap between fourier feature approaches and the full GPs.
> >
> > References:
> >
> > Kakade and Ng, 2005, Online Bounds for Bayesian Algorithms
> >
> > Delbridge et al, 2020, Randomly Projected Additive Gaussian Processes
> >
> > Liu et al, 2022, Incremental Ensemble Gaussian Processes

---

> ### Author Response · Authors · 2024-03-25
> **Response to Reviewer j5fF [3/3]**
>
> > - Basic baselines that would be nice to include for the learning curves are the performance of non-online models trained on various amounts of data.
>
> Because the recursions in DOEBE are exact, the filtered solution of the $m$th expert at time $t$ is exactly the same as the corresponding non-online model, given that they have the same hyperparameters. The regret bounds now included in the manuscript then bound the difference in log-likelihood at time $t$ between an OEBE and the final experts.
>
> > - To alleviate my concern that the ensembles are not necessarily better than a single model, it would be good to study M = 1, 2, 3, …, 5 (or so) on the kuka dataset.
>
> We would like to point the reviewer to Experiment 2; here, an ensemble with two HSGPs (one dynamic and one static) performs radically better than one of its components (the static model). We would also like to remind the reviewer of the online regret bounds included in Section 3.5 of the revised version of the manuscript, which highlights how the ensemble in some sense "hedges" performance to be no worse than its best member.
>
> **Code**
> We very much appreciate your interest and brief review of the code. We respond to your comments regarding the code below.
>
> > - L72 of src/DOEBE/sdoebe.py: should `final_log_w, log_ws` be the same vector?
>
> The array `log_ws` contains the stacked intermediate values of `log_w`, while `final_log_ws` contains only the final value of `log_w` -- see the [lax.scan documentation](https://jax.readthedocs.io/en/latest/_autosummary/jax.lax.scan.html]).
>
> > L118 of src/DOEBE/doebe.py: why is the svd of the pseudo-inverse necessary? Shouldn’t it just be something like (...)
>
> Thank you for this improvement to the code! Indeed, the version of the code at submission essentially computes the SVD twice, and an approach similar to yours is more computationally efficient. The only correction to your suggestion is that if $A = U S V^*,$ then $A^+ = V S U^*$, so $V$ should be used when computing `inv_hess_factor` instead of $U$. This is implemented in the current version of the code as
> ```
> hess_matrix = jnp.block(hess)
> _, s, vh = jnp.linalg.svd(hess_matrix)
> inv_hess_factor = vh.T @ jnp.diag(1 / jnp.clip(jnp.sqrt(s), a_min=1e-16))
> ```
>
>
> **References**
> \[1\] Binois, M., & Wycoff, N. (2022). A survey on high-dimensional Gaussian process modeling with application to Bayesian optimization. ACM Transactions on Evolutionary Learning and Optimization, 2(2), 1-26.

---

> > ### Comment · Reviewer_j5fF · 2024-04-06
> > **Comments**
> >
> > > Because the recursions in DOEBE are exact...
> >
> > Thanks for the clarification!
> >
> > > Experiment 2:
> >
> > Indeed, thanks for the pointer. It still would be good to study how many pieces are in the ensemble (is two required for good performance or do we need all five?)
> >
> > code: Thanks for the clarification.

---

### Review · Reviewer_AyHk · 2024-03-20

**Summary Of Contributions:**

The authors provide a thorough survey of Bayesian methods based on basis expansions including discussions of popular basis choices, online inference, and the ensemble of models using Bayesian model averaging (BMA), and empirical analyses are conducted to support their claims. They further introduce a method to combine static and dynamic models within the BMA framework. The proposed method is compared with other baselines through experiments.

**Audience:**

Yes

**Broader Impact Concerns:**

I have no concerns about the broader impacts and the ethical implications.

**Claims And Evidence:**

Yes

**Requested Changes:**

See the weaknesses & questions above.

**Strengths And Weaknesses:**

Strengths:
1. The authors provide a comprehensive discussion of Bayesian inference based on basis expansions. Quite a few different methods in the literature are properly introduced and compared, and the paper is easy to follow.
2. The method is proposed following clear motivation: avoiding numerical collapse in BMA due to underflow, and Fig. 3 supports why this issue could be problematic.
3. The method is simple and effective according to the experiments.

Weaknesses & questions:
1. What is the benefit of adding a Random Walk on the parameters? The dynamic model in the form of Eq. 26 seems to just make the posterior distribution more conservative. Is there any principled justification for this?
2. How to select $\sigma_{rw}^{(m)}$, and $\delta$ in $Q$? Are they hard to tune?

---

> ### Author Response · Authors · 2024-03-25
> **Response to Reviewer AyHk**
>
> Thank you for your helpful comments on improving our manuscript. We've made several changes to the manuscript based on your comments, which we outline below.
>
> **Weaknesses**
> > 1. What is the benefit of adding a Random Walk on the parameters? The dynamic model in the form of Eq. 26 seems to just make the posterior distribution more conservative. Is there any principled justification for this?
>
> This is an excellent question, and we have added comments in the manuscript similar to the explanation below. We believe a formal justification is most clear in the state-space formulation; in settings where some concept drift is present but unknown, a useful model (barring further knowledge of how this drift occurs) is a random walk in parameter space. Indeed, one may view these as "more conservative" estimates, which is useful in the following sense: in recursive estimation, where we use the posterior at time $t$ for the prior at time $t+1$, a more conservative posterior at time $t$ corresponds exactly to a weaker prior at time $t+1$, which allows for quicker adaptation or "forgetting."
>
> > 2. How to select $\sigma_{rw}^{(m)}$, and $\delta$ in $Q$? Are they hard to tune?
>
> In the current experiments, we choose the value of $\sigma_\text{rw}^{(m)}$ used by the authors of the IE-GP paper, which seems to work well. One benefit of the E-DOEBE model is that it allows for several choices of $\sigma_\text{rw}^{(m)}$ to be chosen and ensembled together. We made a comment regarding this in passing, but have emphasized this in the revised version.
>
> As for $\delta$, we did not tune this at all in our experiments. We performed additional experiments to investigate this, now presented in Appendix F, and found that performance varied with $\delta$, but not by a terribly large amount; choosing $\delta \simeq 0.05$ typically provides good results.

---

### Author Response · Authors · 2024-03-25
**General Comment**

We would like to thank all of the reviewers for their helpful thoughts and comments. We have responded to each reviewer individually and uploaded a revised version of the manuscript. The main changes in the revised version include:
- A section discussing possible methods to incorporate non-Gaussian likelihoods, including the Laplace approximation used in IE-GPs for classification (Section 3.4).
- A section discussing online regret bounds (Section 3.5). The result is essentially the same as for IE-GPs, as the corresponding proof does not use the assumption that the experts are RFF GPs other than for the norm of $\phi^{(m)}(x_t)$. Nonetheless, we provide a proof in a new appendix (Appendix A) for technical completeness and notational consistency.
- We include new experiments in the Appendices for (1) additive HSGPs vs RFF GPs on some UCI benchmark datasets and (2) the sensitivity of the E-DOEBE method to the hyperparameter $\delta$.
- Various minor improvements to grammar, phrasing, and technical presentation. This includes re-ordering the Appendices to match the order in which they are referred to in the main text.

---

### Author Response · Authors · 2024-04-11
**General Comment**

We would like to once again thank the reviewers for their helpful comments and engagement during the discussion period. We have uploaded an updated version of the manuscript including a simple classification experiment (Appendix A), as suggested by reviewer j5fF. We also added a brief paragraph in Section 5.4 pointing the reader to the experiments in appendices.

---

### Decision · Action_Editor_ZmYn · 2024-04-15

**Recommendation:** Accept with minor revision

**Comment:**

As mentioned above, a few points in the presentation could be clarified, hence the minor revision, but otherwise the paper seems to be well-written and technically solid enough to warrant acceptance.

**Audience:**

The reviewers agreed that the topic of ensembling GPs, while not particularly novel, is still interesting enough for the TMLR audience.

**Claims And Evidence:**

The reviewers agreed that most of the claims in the paper are backed by sufficient evidence. However, Reviewer j5fF felt that some of the explanations in the paper (e.g., regarding MMSE estimates, classification experiments, and proofs) could be made even more clear. I would thus encourage the authors to address those comments as a minor revision.